# Sound Corporate Governance and Financial Performance: Is There a Link? Evidence from Manufacturing Companies in South Africa, Nigeria, and Ghana

**Leviticus Mensah * and Murad Abdurahman Bein**

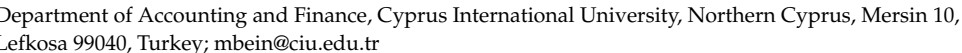

Department of Accounting and Finance, Cyprus International University, Northern Cyprus, Mersin 10, Lefkosa 99040, Turkey; mbein@ciu.edu.tr
* Correspondence: 22000827@student.ciu.edu.tr; Tel.: +233233-24342616

**Abstract:** The study aimed to compare the effect of sound corporate governance on manufacturing companies in South Africa, Nigeria, and Ghana on financial performance. The study used a purposive sampling method to select 60 manufacturing companies, of which twenty-nine (29) were from South Africa, 17 were from Nigeria, and 14 were from Ghana. The study employed GMM and FMOLS to estimate the effect of corporate governance on the firm's financial performance. According to the study, South Africa has the longest average board tenure at 7.85 years, followed by Nigeria at 4.7 years and Ghana at 3.9 years. The average board tenure was found to have a positive and statistically significant effect on the return on invested capital (ROIC) of the firms in South Africa and Ghana, and a positive and statistically insignificant effect was found for the firms in Nigeria. The study indicated that the firms in South Africa have the highest percentage of female directors at 24.26%, followed by Ghana at 17.8% and Nigeria at 17.3%. The study showed that female representation on the corporate board has a positive and statistically significant effect on all firms' return on net operating assets (RONOA). The study provides policy implications for shareholders, boards of directors, and other stakeholders by enabling them to build confidence in the corporate governance structure of manufacturing companies in the three countries.

**Keywords:** board tenure; corporate governance; Ghana; financial performance; Nigeria; South Africa



## 1. Introduction

Manufacturing companies in Ghana, Nigeria, and South Africa are among Africa's fastest-growing manufacturing industries. The manufacturing industry acts as a catalyst for developing related industries such as services and achieving targeted goals such as economic independence and job creation [1]. The potential for rapid economic expansion offered by the manufacturing sector cannot be overlooked. The manufacturing sector in Africa adds more value to GDP than any other industry [2]. Investment in greater value-added, export-led manufacturing is crucial for Africa's industrialization and growth [3]. According to a World Bank indicator for 2021, in Sub-Saharan Africa's total GDP, the manufacturing firms in Nigeria contributed 5%, South Africa contributed 15%, and Ghana contributed 12%. The findings indicate that a strong manufacturing sector is one of the factors driving development in Africa.

High firm valuations result from the company's substantial success, which draws in investors and other stakeholders [4]. On the other hand, poor performance might harm a company's stock value. The effectiveness and efficiency of the firm's actions during a specific period are reflected in the company's performance, which is also the consequence of the company's formal efforts [5]. A company's track record speaks volumes about its credibility in the eyes of financiers, clients, and the public. Likewise, a company's financial results are used to judge its success and serve as a foundation for making decisions. Deciding whether to keep or sell an investment, an investor assesses how his/her money is faring

in the financial market. Manufacturing enterprises perform well and attract more investors from all over the world when there exists a solid corporate governance framework where the board of directors plays a vital role in supervising the management of the business [6]. Organizations should have a corporate governance structure in place [7] because it allows for ownership and control to be kept separate from management. This increases the likelihood that professionals with the necessary expertise and in-depth business knowledge will run the company. According to the agency theory, managers frequently seek their self-interest at the company's and its stockholders' expense [8].

One of the issues confronting manufacturing companies in Africa is weak corporate governance, which makes it impossible to attract investors [9]. Weak corporate governance has affected Africa and even developed countries such as the USA, Germany, France, and the UK [10]. Global business fraud, scandals, and failures can be directly attributed to poor corporate governance. Companies such as Parmalat in Italy, WorldCom, and Enron in the USA, Tyco in the Republic of Ireland, Sanlu in China, and Tata Mistry and Satyam Computers in India are all victims of fraud scandals that have resulted in major losses for these companies' stockholders [11]. In Ghana, weak corporate governance led to the collapse of these companies (Capital Bank, DKM, Ghana Airway Limited, Jesta Motors, Juapong Textiles Limited, Microfinance, and UT Bank), where eventually the investors lost all their wealth [12]. In Nigeria, these companies (Oceanic Bank, Cadbury, Gulf Bank of Nigeria Plc, Societe General Bank of Nigeria, Metropolitan Bank Limited, Lead Bank Plc, Assurance Bank Nigeria Limited, and Hallmark Bank Plc) collapsed, and millions of dollars of investor funds were not refunded due to poor corporate governance structures [13]. In South Africa, futile corporate governance led to reduced shareholder value or even to the collapse of companies, including Unifer, Masterbond, Saambou Bank, CNA, Tollgate, Leisurene, and Fidentia [14]. One of the ways to restore investor confidence is by instituting more robust corporate governance.

Advanced countries such as Germany, the United States, and the United Kingdom have taken adequate measures to raise corporate governance standards and prevent such failures. Several African countries, notably South Africa, Nigeria, and Ghana have instituted measures in the same regard. When shareholders appoint a board of directors to represent them and give them the authority to oversee, assess, and supervise management's actions [15,16]. The board of directors, which acts as the company's owners' representative, is responsible for the company's successful management. To guarantee that management's activities align with the owner's interests, the board must implement control measures [17]. The board of directors is primarily responsible for appointing and firing the CEO and other senior executives, determining management salaries, monitoring manager performance, and assessing the company's overall performance. The law requires directors to support management initiatives, assess management performance, and decide on management compensation to maximize shareholders' value [18].

The study seeks to answer the question, "Does corporate governance affect ROIC, RONOA, and Tobin's Q as a measure of the financial performance of the firms?" We seek to contribute to the literature in the following ways: First, find the effect of corporate governance on ROIC as a measure of financial performance. ROIC measures firms' performance regarding the board's access to capital for the organization. Excellent corporate governance makes access to funds (both equity and debt capital) more accessible, which boosts the company's long-term competitiveness [19]. More investors are willing to supply capital at lower costs to companies that promote strong corporate governance practices and implement the highest governance standards since the inherent risk of investing in shares can be minimized to the greatest extent possible. The results should determine how sound corporate governance affects the various sources of capital the firms use to manage the corporation, including capital obtained from investors, and would enable the investors to have confidence in the corporation. Secondly, we contribute to the literature by finding the effect of corporate governance on RONOA as a measure of firm operating performance. Investors look at the operating performance of the corporation. The RONOA

measures firms' profit from using their operating assets, excluding any investment made. The corporation can invest and obtain a higher return on their investment, which would be reported to shareholders; however, the shareholders cannot access the efficiency of the management by relying on the return on assets, which includes the investment component, which is not the result of the management's hard work. Thirdly, we contribute to the literature by comparing the effects of corporate governance in South Africa, Nigeria, and Ghana to find out which corporate governance measures installed in the manufacturing sector contribute significantly to financial performance. Lastly, we contribute literature by finding the effect of corporate governance on Tobin's Q, which measures the market performance of the firms. Tobin's Q findings would address these issues by encouraging efficient resource usage and simultaneously demanding accountability from those entrusted with those resources. One of the objectives of a corporation is to grow, expand, survive, and contribute to the sustainable development of the country in which it operates through employment, the payment of taxes, and others. Poor corporate governance structures will lead to the corporation's collapse, where the objective would not be achieved, but having robust corporate governance is a way to ensure the corporation's growth and survival by contributing to sustainable growth and development in the jurisdiction of operations.

Management monitoring, transparency, protection of all shareholders' interests, control, and independent examination and approval of the corporation's strategy and material business plans and actions by the board are the cornerstones of sound corporate governance [20]. These elements help the company maximize shareholder value by fostering superior, long-term performance and value generation. Stakeholders can shield themselves from the opportunistic behavior of business management by using the processes of corporate governance [21]. According to corporate governance principles, businesses should balance the needs of their owners and that of all other stakeholders. The rest of the study is organized as follows: literature review, material and methods, empirical analysis and discussion, conclusions, managerial implications, limitations of the study, and direction for future studies.

## 2. Literature Review

There are many corporate governance theories, but the ones that support the studies are discussed below.

### 2.1. Theories of Corporate Governance

2.1.1. Agency Theory

Managerial decisions in today's widely owned corporations deviate from what is necessary to maximize shareholder returns. According to agency theory, managers are agents, and owners are principals; an agency loss occurs when owners' returns on their residual claims are lower than if the owners themselves exercised direct management of the firm [22]. Mechanisms for minimizing agency losses are outlined in the agency theory. They include incentive programs for managers that provide financial rewards for maximizing shareholder value. Such strategies usually involve top executives buying shares, sometimes at a discount, to align their financial interests with shareholders [23]. Several similar systems relate CEO salary and benefits to shareholder returns and defer a portion of executive income to the future to promote long-term value maximization and discourage short-term executive activity that undermines business value [24]. When the managers of an organization have different goals from the shareholders and the management withholds essential information about manufacturing companies from the shareholders, this may impact the business's financial performance. The shareholders of the manufacturing companies may not be willing to contribute additional funds to the corporation when they discover that their objectives and the information that they were receiving differed from those of the managers.

### 2.1.2. Stewardship Theory

The stewardship theory of corporate governance downplays the possibility of tensions between business management and shareholders and favors an insider-dominated board of directors [25]. The idea presupposes that managers are trustworthy and value their reputations highly. As a result, the market for highly regarded managers, who command higher salaries, is the essential tool to regulate behavior [26]. Financial reporting, disclosure, and auditing are still essential, but there is a fundamental assumption that these practices are necessary to confirm management's inherent integrity. One of the ways the managers account to the corporation's shareholders is through annual general meetings. If the managers do not account to the shareholders how effectively they used the corporation's assets to generate enough earnings, represented by ROA and ROE, then the shareholders, through the board of directors, may not approve any assets the managers may need in the future to run the corporation, which could affect the corporation's financial performance.

### 2.1.3. Stakeholder Theory

According to stakeholder theory, a company or organization must serve multiple constituencies. According to stakeholder theory, everyone who impacts the company or its operations in any way is a stakeholder, including environmental organizations, clients, employees, local communities, suppliers, governmental organizations, and more [27]. According to stakeholder theory, organizations and businesses will be more successful in the long run if they prioritize their constituents' needs. According to shareholder theory, a company's primary purpose should be to advance the interests of its shareholders. Shareholder theory translates to a making-more-profit-at-any-cost business philosophy, because shareholders are most concerned with financial growth [28]. The success of the corporation depends on its primary and secondary stakeholders. The corporation's failure to meet or address its stakeholders' needs may lead to its collapse in the future. For example, suppose manufacturing companies fail to honor their tax obligations to the government. In that case, it may affect their operations due to the frustration they would face from the government, or the government could collapse the corporation. Therefore, stakeholder theory is vital to a manufacturing company's corporate governance and financial performance.

### 2.1.4. Resource Dependency Theory

According to resource dependency theory [29], the board of directors plays a pivotal role in securing the necessary external resources for an organization. Theorists of resource dependency emphasize the importance of having third-party representatives oversee the acquisition of essential resources. The availability of resources improves organizational effectiveness, productivity, and survival. The directors give the company credibility [30] and provide information, expertise, and connections to important stakeholders such as government policymakers, consumers, and business partners. The smooth running of a corporation depends on its resource. Failure to supply the corporation with resources in terms of processing inputs could affect the corporation's performance. Every successful corporation depends on its inputs, such as human and capital resources.

### *2.2. Corporate Governance Regulations in South Africa, Nigeria, and Ghana*

### 2.2.1. Corporate Governance of South Africa

South Africa uses a mix of common law, legislation, soft law, and market regulation to govern its corporations [31]. South Africa collaborates with its G20 peers to establish rules for the global regulation of financial markets. Its leadership in spreading integrated reporting worldwide has informed its approach to corporate governance, commonly referred to as the King Code; the South African Corporate Governance Code was primarily authored by the retired chairman of the International Integrated Reporting Council and the Global Reporting Initiative. Companies must adhere to various benchmark requirements set forth by the King Code in their public disclosures and internal operations [32]. A general guide on how businesses should operate can help standardize sectors and attract foreign

investment. All firms doing business in South Africa must adhere to some rudimentary norms of conduct, as this is one of the King Code's primary functions [33,34]. Shareholders and the general public can hold a firm accountable if it disobeys a King Code guideline.

### 2.2.2. Corporate Governance in Nigeria

A statutory framework and subsidiary legislation issued by the necessary regulatory authorities make up the Nigerian corporate governance regime [35]. There are two types of laws: general laws and industry-specific legislation. The sector-specific rules only apply to businesses that operate within their particular sector or industry, while the general laws apply to all entities formed in Nigeria [36]. The general laws are as follows:

The Act Concerning Companies and Allied Matters; the Corporate Affairs Commission (CAC) enforces the CAMA. The Investments and Securities Act of 2007 (ISA) established the Securities and Exchange Commission (SEC) as its regulatory body. This organization is responsible for enforcing the Financial Reporting Council of Nigeria Act 2011 (FRCN Act) provisions.

The Banks and Other Financial Institutions Act (BOFIA) and the Insurance Act are two examples of industry-specific laws (IA). The CAMA is the primary law that defines the broad parameters of the Nigerian corporate governance regime [36]. Private companies, limited by shares or guarantees, limitless corporations, and public companies, which shares can only limit, are all described as possible forms of incorporation. The CAMA also specifies several corporate bodies' roles, responsibilities, and authority, along with the company's governance and management systems, and the management requirements [37]. However, the ISA lays forth the Nigerian securities market's rules and regulations. It specifies the liquidity requirements and the operational norms for market participants, operators, and stakeholders.

### 2.2.3. Corporate Governance in Ghana

Statutory law, subsidiary legislation, regulatory guidelines, and directives comprise most of Ghana's corporate governance framework for publicly traded companies [38]. The Corporate Governance Code is not legally binding; it serves as a guideline against which publicly traded companies and other organizations active in the securities market can be measured. Non-compliance with the strict mandatory rules that some financial services authorities have established for the governance structures and control systems of regulated organizations could have severe consequences for the companies' licenses. For instance, the National Insurance Commission has established regulations for both life and non-life insurers regarding governance and risk management. The comprehensive framework outlines the minimal standards for internal control systems and corporate governance frameworks that insurers must follow. This includes the audit and risk management functions, the board's makeup, and the roles of any relevant committees [39].

### 2.3. Empirical Studies

There have been studies on the effect of corporate governance on financial performance in South Africa. Dzingai and Fakoya [16] used companies in the mining sector listed between 2010 and 2015 and found that board size has a negative and statistically insignificant effect on ROE, while board independence has a positive and statistically significant effect on ROE. Toshiba et al. [17] found that corporate attributes had a positive and statistically significant effect on the financial performance of manufacturing companies during and after the 2008 financial crisis. Karaye and Büyükkara [18] used stock market data for companies listed in Johannesburg between 2013 and 2019 and found that the corporate governance index has a positive and statistically significant relationship with ROA. In Nigeria, Urhoghide and Omolaye [40] used 14 oil and gas companies listed in the Nigerian Stock Market between 2008 and 2015. They found that board size and female representation on the board have positive and statistically significant effects on financial performance. Yusuf, Bambale, and Abdullahi [41] used eight financial institutions listed

in the Nigerian stock market in 2015 and found that board size and composition have positive and statistically insignificant effects on financial performance. In Ghana, Musah, and Adutwumwaa [42] used 10 rural banks' years of data between 2010 and 2019 and found that board size and CEO duality have positive and statistically insignificant effects on ROA and ROE. Boachie [43] used 23 financial institutions' data sets between 2006 and 2018, and the study showed that CEO duality and non-executive directors have a positive and statistically significant relationship with financial performance. Aside from South Africa, Nigeria, and Ghana, similar studies have been conducted across the globe. Hakimah et al. [44] used 50 samples of Indonesian SMEs between 2013 and 2017 and found that board size and female representation have a positive and statistically significant effect on ROA. Kyere and Ausloos [45] used 252 firms listed on the London Stock Exchange in the year 2014 and found that board size, independent directors, and CEO duality have a positive and statistically significant relationship with ROA, while a positive and statistically insignificant relationship was found using Tobin's Q. Al-Homaidi [46] used 30 banks between 2013 and 2016 listed on the Bombay Stock Exchange and found a statistically significant relationship with ROA. These studies used ROA, ROE, EPS, and Tobin's Q to measure financial performance. However, the effect of corporate governance on financial performance, where the return on invested capital (ROIC) and return on net operating assets (RONOA) measure financial and operating performance, has not been well explored in the literature. Moreover, comparative studies have not been carried out on these three countries, South Africa, Ghana, and Nigeria, to find out corporate governance's effect on manufacturing firms' financial performance.

## 3. Materials and Methods

### 3.1. Sample and Data

The study selected three countries: South Africa, Nigeria, and Ghana. The three countries were chosen for the study based on manufacturing companies' data availability and adoption of corporate governance by the manufacturing companies listed on the stock markets. The study used companies in the manufacturing sector instead of those in the service sector. In Ghana, manufacturing contributes significantly to the country's economic growth by offering employment, paying taxes, and performing corporate social responsibilities [47]. In Nigeria, manufacturing companies contribute to the social and economic development of the country [48]. South Africa is the leading and most influential in development, contributing significantly to the gross domestic product [49]. The study used a purposive sampling technique to select 29 manufacturing companies in South Africa between 2011 and 2020, giving 290 firm years of observation; in Nigeria, the study selected 17 manufacturing companies between 2011 and 2020, giving 170 firm years of observation; and in Ghana, 14 manufacturing companies between 2011 and 2020 were selected, giving 140 firm years of observation. The data used for the study from South Africa were downloaded from Thomson Reuters Eikon DataStream, and that of Ghana and Nigeria were extracted from their annual reports published. Amin and Cek [50] used data from Thomson Reuters Eikon to examine the effect of capital structure deviation from the golden ratio on the financial performance of firms in the UK and French stock markets. Boachie [43], Musah and Adutwumwaa [42] and Arhinful, Mensah Owusu-Sarfo [51] also extracted data from the annual report of firms listed in the Ghana Stock Market to find the effect of the corporate governance system on financial performance, and in Nigeria, Urhoghide, and Omolaye [40] and Yusuf, Bambale, and Abdullahi [41] also used data from firms listed on the Nigerian Stock Market for the same purpose. The companies selected in South Africa were listed on the Johannesburg Stock Exchange; the firms in Nigeria were listed on the Nigerian stock market; and the firms in Ghana were listed on the Ghana stock market.

*3.2. Dependent and Independent Variables and Hypotheses Development*

The study used three dependent variables and eight independent variables. All the dependent and independent variables are summarized in Table 1, including their measurement.

**Table 1.** Summary of study variable and measurement.

| Variables | Measurement |
|---|---|
| **Dependent variable:** | |
| Return on invested capital | Net operating profit after tax/(debt + equity) $\times$ 100 |
| Return on net operating assets | Net operating profit after tax/net operating assets. Net operating asset = operating assets $-$ operating liabilities |
| Tobin's Q | Total Market Value of Firm/total asset value of the firm |
| **Independent variable:** | |
| Board tenure | The number of years the firm's directors have served on the board/total directors serving on the board |
| Number of board meetings | Total number of meetings in a year |
| Board size | Total number of board members on the corporate board |
| CEO duality | The dummy variable if the CEO is a board chair marked '1' or otherwise '0' |
| Non-executive director | Total number of non-executive directors/total number of board members |
| Female representation | Total number of females/total number of board members |
| Board ownership | Total shareholding by the directors/total outstanding shares |
| Board compensation | Log of compensation to board members |

3.2.1. Dependent Variables

The study used three independent variables: return on invested capital (ROIC), return on net operating assets (RONOA), and Tobin's Q.

The return on invested capital (ROIC) is the percentage of profitability that a business achieves using its invested equity and debt capital [52]. Because the continual creation of positive value is seen favorably as a necessary trait of a great business [53,54], ROIC is widely employed to measure the efficiency with which capital is invested. It measures how profitable a company is relative to the money invested in its operations.

The gap between operating and operational liabilities is the return on net operating assets (RONOA) [55]. Assets used by a company to make money and run its day-to-day operations are known as net operating assets. Business operations and financial activities (investments) are the two primary sources of business income [56]. The potential profits from corporate activities would be of interest to investors. The amount of money made from core business activities can therefore be determined by the return on net operating assets. Return on Net Operating Assets measures the profit from a company's revenue-generating assets. It is a vital sign of how efficiently a business turns its assets into cash [57]. Companies with a higher RNOA typically attract more interest from investors.

Tobin's Q ratio is calculated by dividing the firm's market value by its asset value [58]. Suppose the market value of a company is higher than its replacement cost. In that case, the company is overpriced according to Tobin's Q ratio. Suppose a company has a higher ROE than its cost of capital. In that case, it will likely inspire other entrepreneurs to launch copycat enterprises to share in the profits. Suppose a company's Tobin's Q ratio is less than 1.0. In that case, it is considered undervalued, and investors are more likely to be interested in investing in such a company [59].

3.2.2. Independent Variables and Hypothesis Development

The study used eight corporate governance mechanisms: board tenure, number of board meetings, board size, CEO duality, non-executive directors, female representation, board ownership, and board compensation.

Board tenure

The board tenure year describes the period from the date of one annual shareholder meeting to the date of the next annual shareholder meeting [60]. The average tenure of the board's directors affects both the depth of its familiarity with the business and its capacity for impartial decision making. On the one hand, the worth of a company might rise as employees gain experience working for that particular company. Directors who have been on the board for a while are likelier to be friends with management and less likely to question their judgments [61]. Nevertheless, directors with more experience and familiarity with the company's operations are better positioned to offer recommendations [62]. Mentorship programs between senior and junior directors have been shown to speed up the learning curve for new directors.

Extended board tenure is connected with poor firm performance and is a source of concern for shareholders [63]. On the other hand, too much intimacy between the board and management might lead to a loss of independence. They must be more resistant to learning new things or making the necessary strategic adjustments. Seasoned board members warm up to managers over time. As boards lose their independence, they become less effective at monitoring management and less able to protect the company's value. Long-serving board members are more likely to have a fiduciary relationship with the company, to be associated with managers from the start of their service, and to hold positions of greater influence and equity in the company [62].

**Hypothesis 1 (H1).** *Board tenure statistically significantly affects manufacturing firms' financial performance.*

Number of board meetings

The frequency of board meetings per year can be used to gauge the level of board activity [64]. According to the corporate governance perspective, the board should hold regular meetings to address any issues. In board meetings, the board's supervisory responsibilities are carried out. The frequency of board meetings can be used to gauge how well the board is carrying out its duties and obligations [65]. Suppose board decisions successfully reduce agency costs and conflicts of interest. In that case, more frequent board meetings will increase the principal value [45]. The directors can assess the effectiveness of existing plans and the management team's work when the board frequently meets [8]. Board meetings have a major positive effect on a company's performance. The board meeting is another crucial, value-relevant board characteristic. Board meetings have been shown to improve decision making in certain studies, but others have suggested that the time and money spent on them make them ineffective [66].

Since most board meetings are spent debating the management report and other routine business, there is no correlation between the frequency of board meetings and the maximization of shareholder value [67]. The monitoring function will need to be improved, agency costs will increase, and corporate performance will suffer if independent directors are not actively running the company [68]. At board meetings, costs such as administrative time, director compensation, and travel expenditures mount up and hurt a company's bottom line. Members may need more time to address urgent issues at infrequent board sessions [69].

**Hypothesis 2 (H2).** *The number of board meetings statistically significantly affects the manufacturing firm's financial performance.*

Board size

It has been stated that businesses are less likely to make drastic decisions when board size increases, reducing corporate performance variability [70]. Due to the increased inspection and monitoring function the board provided, businesses with larger boards produced better financial reporting [71]. Larger boards may facilitate an efficient monitoring function from the agency's perspective by deploying more seasoned directors and decreasing CEO dominance [72].

When too many directors are on the board, it is difficult for them to keep an eye on the management [73]. The number of directors on a business's board, the highest decision-making body, has long been a key indicator of how well the company performs.

**Hypothesis 3 (H3).** *Board size statistically significantly affects the manufacturing firm's financial performance.*

CEO duality

CEO duality is one of the board control methods of CG approaches. It is referred to as having one individual fulfill two positions when a company's CEO also serves as the chair of the board of directors [74]. Different experts have divergent opinions on the connection between CEO dualism and business performance. According to agency theory, duality can make it harder for the board to carry out its oversight role, which, in turn, can exacerbate current agency problems and increase production that needs to be revised [75]. Dual leadership reduces board independence and enhances CEO entrenchment.

Due to the trust placed in one person to carry out two distinct but connected jobs, that individual may discover that the demands of both positions conflict with their time as CEO. The CEO is in charge of implementing the company's strategies and monitoring and evaluating management's performance. It has been argued that keeping the board's leadership distinct from the firm's day-to-day operations is an advantage, which is why the CEO's responsibilities should be kept separate [76]. According to the stewardship theory of corporate governance, managers have the best interests of the firm at heart and act as responsible stewards of its assets [77]. Suppose the CEO and the chairman are the same people. In that case, there should be no conflict of interest between shareholders and management, according to the stewardship hypothesis. Instead, having two CEOs would strengthen the company's leadership and make it easier to set clear goals [78].

**Hypothesis 4 (H4).** *The CEO duality statistically significantly affects the manufacturing firm's financial performance.*

Non-executive directors

A governance strategy hotly debated in the literature is the appointment of independent non-executive directors to a company's board to oversee management on behalf of shareholders [79]. Having a substantial number of independent non-executive directors on the board can boost the efficiency of controls. Including non-executive directors on the corporate board is advantageous because it keeps the board's ability to exercise control impartial and lowers the likelihood of conflicts of interest between the principal and the agent. To protects minority shareholders and other interested parties, independent non-executive directors also ensure balanced decision making [68]. It has been concluded that increased numbers of independent non-executive members on boards can enhance the credibility of financial reports and reduce the need for financial restatements [80].

It has been asserted that organizations that empower outside directors have better monitoring and, consequently, better performance [81]. It is argued that independent non-executive directors have a negative effect on firm performance because they do not have a sufficient understanding of the firm's day-to-day business activities due to their lack of access to information about the internal business of firms and their relationship with internal management [82].

**Hypothesis 5 (H5).** *Non-executive directors have a statistically significant effect on the manufacturing firm's financial performance.*

Female representation

In nations with weaker external processes, the gender balance of a board may have an impact on the quality of the controlling function and corporate performance [83]. According to stakeholder theory, having more women on the board means paying more attention to

and focusing on the demands and interests of stakeholders, which, in turn, improves board performance and boosts company success [84]. According to agency theory, having female board members reduces the power of male board members. Women are better monitors than men and can juggle multiple jobs at once.

Better monitoring, which will affect business values, has been linked to the participation of women on boards. The board of directors would function more efficiently with more female directors. By utilizing their skills in human resource management, legal, communication, and public relations, female directors are more likely than their male counterparts to assist the company's administration [85]. Female directors add unique and priceless relationships to their boards with their diverse experience, strengths, and knowledge in observation and perceptiveness [86].

**Hypothesis 6 (H6).** *Female representation on the corporate board statistically significantly affects the manufacturing firm's financial performance.*

Board ownership

It is commonly accepted that ownership structure is the primary factor influencing the effectiveness of CG mechanisms [87]. Agency theory states that concentrated shareholders may successfully manage corporate operating management, resolve information issues, and save agency expenses, improving company performance [88]. According to agency theory, the agency problem and conflicts of interest are reduced when directors own shares and are actively involved in the company's day-to-day operations [89]. According to agency theory, institutional share ownership promotes monitoring since institutional shareholders often hold most of the shares and exert significant control over management, which reduces the agency problem [90].

It is generally accepted that a concentrated ownership structure reduces agency costs by encouraging major shareholders to oversee management. To maximize their interests, shareholders would be able to influence decision making through higher ownership concentrations. There is further evidence that concentrated stockholders might boost their benefits at the expense of smaller shareholders [91,92].

**Hypothesis 7 (H7).** *Board ownership statistically significantly affects the manufacturing firm's financial performance.*

Board Compensation

According to agency theory, corporate boards have the authority to direct management's actions in the best interests of shareholders in arm's-length transactions. Directors' pay plans should be designed to provide adequate incentives to increase shareholder value and mitigate the moral hazard risk that results from the separation of ownership and control [93]. High stock returns indicate managerial effort; thus, pay should be linked to stock-based performance metrics [94]. This is in line with the wishes of shareholders. The shareholders vote annually to approve the directors' compensation packages. Agency theorists generally agree that the most effective means of reducing agency difficulties are for the board of directors to keep a close eye on management and offer board members financial incentives. [95]

Making the board of directors' remuneration responsive to business success through share ownership and bonuses [96] is one strategy to reduce the potential for a conflict of interest between management and shareholders. The goal of establishing compensation packages for board members is to make their pay sensitive to the firm's performance. Compensation for board members can be a significant incentive for them to participate in efforts to boost the company's performance. Directors' compensation is a market mechanism for tying managerial decision making to the development of shareholder value, hence discouraging excessive rent extraction by executives [97]. It is recommended that the compensation committee assumes complete responsibility for the compensation process, policies, and procedures [98].

**Hypothesis 8 (H8).** *Board compensation has a statistically significant effect on the manufacturing firm's financial performance.*

The dependent variable measures the financial performance of the firms, and the independent variables are proxies for corporate governance.

*3.3. Model Specification*

To determine the effect of corporate governance on the financial performance of manufacturing firms in South Africa, Nigeria, and Ghana, the model below was used.

$$FP\,I,t = \beta 0\,I,t + \beta 1\,BTE\,I,t + \beta 2\,NBM\,I,t + \beta 3\,BSZ\,I,t + \beta 4\,CSY\,I,t + \beta 5\,NXD\,I,t + \beta 6\,FPN\,I,t$$
$$+ \beta 7\,BOP\,I,t + \beta 8\,I,t\,BCN\,I,t + \varepsilon\,I,t \tag{1}$$

where "*FP*" is the dependent variable, which is measured by ROIC, RONOA, and TOBB. "*BTE*" denotes board tenure; "*NBM*" denotes the number of board meetings in a year; "*BSZ*" denotes board size; "*CSY*" denotes CEO duality; "*NXD*" denotes non-executive director; "*FPN*" denotes female representation on the corporate board; "*BOP*" denotes board ownership; "*BCN*" denotes board compensation; and "*ε*" denotes the error term.

The study used the generalized method of movement (GMM) and fully modified ordinary least squares (FMOLS) for the empirical analysis. For dealing with autocorrelation and unobservable fixed effects concerns in dynamic panel data, Arellano and Bond [99] argue that system GMM is a superior estimation method. Phillips and Hansen [100] developed the fully modified least squares (FMOLS) regression model to produce the most precise estimates of cointegrating regressions. The least squares are modified to account for serial correlation effects and endogeneity in the regressors due to a cointegrating relationship. Both estimations can deal with autocorrelation and endogeneity, which gives us the reason to select them over the fixed effect model, random effect model, and ordinary least squares model.

**4. Results and Discussion**

*4.1. Descriptive Statistics*

Tables 2–4 show the descriptive statistics for the three countries: South Africa, Nigeria, and Ghana. The firms in Nigeria have the highest ROIC, followed by those in South Africa and Ghana. Regarding RONOA, the firms in Nigeria have the highest RONOA, followed by South Africa and Ghana. The results show that all the firms in the three countries are undervalued since Tobin's Q is less than one. The firms in Ghana have the highest Tobin's Q, followed by Nigeria and South Africa. The board members in South Africa have the longest tenure, followed by Nigeria and Ghana. The board members in South Africa have more experience than those in Nigeria and Ghana. The board members in South Africa meet more often followed by Nigeria and then Ghana. Board size is the number of board members in the corporate room. The firms in South Africa have the largest number of board members compared to those in Nigeria and Ghana. Nigerian businesses have the second-largest number of board members, followed by Ghanaian businesses.

The manufacturing firms in South Africa have more dual CEO roles than those in Nigeria and Ghana. Comparison of CEO duality between Nigeria and Ghana: Nigeria has more dual CEO roles than the firms in Ghana. The firms in Nigeria have a greater percentage of non-executive directors on their corporate boards than those in South Africa and Ghana. However, the firms in South Africa have a greater percentage of non-executive directors than those in Ghana. The firms in South Africa have the highest number of female directors, followed by those in Nigeria and Ghana. The board members in Ghana have a greater percentage of shares in the firms than the manufacturing firms in South Africa and Nigeria. Comparing the shareholdings of the board members between Nigeria and South Africa, the board members in Nigeria have the highest shareholdings. Compared to the board members in the other two countries, Nigerian board members received the highest

compensation. Board members in Ghana received the second-highest compensation, and South African board members received the lowest compensation.

**Table 2.** Descriptive Statistics (South Africa).

| Variable | Obs | Mean | Std. Dev. | Min | Max |
|---|---|---|---|---|---|
| Return on invested capital | 290 | 7.64 | 8.88 | −40.93 | 33.528 |
| Return on net operating assets | 290 | 14.789 | 22.955 | −146.724 | 67.491 |
| Tobin's Q | 290 | 0.011 | 0.051 | 0.001 | 0.366 |
| Board tenure | 290 | 7.852 | 2.566 | 1.96 | 15.96 |
| Number of board meetings | 290 | 5.34 | 2.011 | 0.25 | 20 |
| Board size | 290 | 11.033 | 2.488 | 0.219 | 23 |
| CEO duality | 290 | 0.951 | 0.215 | 0 | 1 |
| Non-executive director | 290 | 72.348 | 10.128 | 5.693 | 90.91 |
| Female representation | 290 | 24.256 | 10.435 | 0 | 58.33 |
| Board ownership | 290 | 0.562 | 0.861 | 0 | 4.83 |
| Board compensation (R) | 290 | 802,396 | 0.474 | 1736 | 26,165,955 |

**Table 3.** Descriptive Statistics (Nigeria).

| Variable | Obs | Mean | Std. Dev. | Min | Max |
|---|---|---|---|---|---|
| Return on invested capital | 170 | 9.427 | 11.833 | −46.185 | 53.959 |
| Return on net operating assets | 170 | 24.052 | 20.593 | 1.378 | 66.209 |
| Tobin's Q | 170 | 0.04 | 0.071 | 0.002 | 0.393 |
| Board tenure | 170 | 4.665 | 1.171 | 2 | 7 |
| Number of board meetings | 170 | 5.141 | 1.188 | 4 | 10 |
| Board size | 170 | 9.894 | 2.767 | 5 | 17 |
| CEO duality | 170 | 0.482 | 0.501 | 0 | 1 |
| Non-executive director | 170 | 74.082 | 11.281 | 43 | 93 |
| Female representation | 170 | 17.306 | 13.112 | 0 | 67 |
| Board ownership | 170 | 1.119 | 2.962 | 0 | 14.48 |
| Board compensation (₦) | 170 | 144,748,057 | 0.999 | 58,683 | 1,478,000,000 |

**Table 4.** Descriptive Statistics (Ghana).

| Variable | Obs | Mean | Std. Dev. | Min | Max |
|---|---|---|---|---|---|
| Return on invested capital | 140 | 1.621 | 12.52 | −40.283 | 30.989 |
| Return on net operating assets | 140 | 4.28 | 34.297 | −70.036 | 46.473 |
| Tobin's Q | 140 | 0.142 | 0.383 | 0.002 | 4.331 |
| Board tenure | 140 | 3.9 | 1.456 | 2 | 7 |
| Number of board meetings | 140 | 4.607 | 0.784 | 4 | 7 |
| Board size | 140 | 8.4 | 1.933 | 5 | 13 |
| CEO duality | 140 | 0.15 | 0.358 | 0 | 1 |
| Non-executive director | 140 | 67.141 | 23.878 | 14.3 | 92 |
| Female representation | 140 | 17.675 | 15.278 | 0 | 60 |
| Board ownership | 140 | 4.965 | 11.163 | 0 | 35 |
| Board compensation (GH¢) | 140 | 7,358,264 | 1.042 | 104 | 437,654,000 |

### 4.2. Matrix Correlation Analysis

Tables 5–7 show the results for the matrix correlation for the three countries: South Africa, Nigeria, and Ghana. The results obtained for the corporate governance variables, which are explanatory, are not correlated with each other since their values from the correlation matrix tables are not greater than 0.70 [101]. The results show that there is no problem of multicollinearity, which, when present, will give a biased estimation. One of the assumptions of multiple linear regression is that the explanatory variable, which is the corporate governance variables (board tenure, number of board meetings, board size,

CEO duality, non-executive directors, female representation, board ownership, and board compensation), should not be strongly correlated with each other [102].

**Table 5.** Matrix of correlations (South Africa).

| Variables | (1) | (2) | (3) | (4) | (5) | (6) | (7) | (8) | (9) | (10) | (11) |
|---|---|---|---|---|---|---|---|---|---|---|---|
| (1) Return on invested capital | 1.000 | | | | | | | | | | |
| (2) Return on net operating assets | 0.039 | 1.000 | | | | | | | | | |
| (3) Tobin's Q | −0.140 | −0.033 | 1.000 | | | | | | | | |
| (4) Board tenure | 0.247 | −0.029 | 0.077 | 1.000 | | | | | | | |
| (5) Number of board meetings | −0.275 | −0.133 | 0.148 | −0.186 | 1.000 | | | | | | |
| (6) Board size | −0.008 | 0.067 | −0.067 | −0.042 | 0.140 | 1.000 | | | | | |
| (7) CEO duality | −0.002 | 0.027 | 0.043 | −0.004 | −0.084 | 0.039 | 1.000 | | | | |
| (8) Non-executive director | −0.029 | −0.120 | 0.170 | −0.235 | 0.260 | 0.056 | −0.105 | 1.000 | | | |
| (9) Female representation | −0.174 | −0.111 | 0.034 | −0.196 | 0.168 | 0.103 | 0.070 | 0.109 | 1.000 | | |
| (10) Board ownership | 0.103 | −0.086 | 0.150 | 0.084 | 0.056 | 0.034 | −0.163 | 0.087 | 0.012 | 1.000 | |
| (11) Board compensation | −0.060 | −0.123 | −0.028 | −0.037 | 0.069 | 0.305 | −0.010 | 0.177 | 0.147 | 0.012 | 1.000 |

**Table 6.** Matrix of correlations (Nigeria).

| Variables | (1) | (2) | (3) | (4) | (5) | (6) | (7) | (8) | (9) | (10) | (11) |
|---|---|---|---|---|---|---|---|---|---|---|---|
| (1) Return on invested capital | 1.000 | | | | | | | | | | |
| (2) Return on net operating assets | −0.022 | 1.000 | | | | | | | | | |
| (3) Tobin's Q | 0.271 | −0.046 | 1.000 | | | | | | | | |
| (4) Board tenure | 0.021 | −0.153 | −0.010 | 1.000 | | | | | | | |
| (5) Number of board meetings | −0.013 | 0.024 | 0.034 | 0.013 | 1.000 | | | | | | |
| (6) Board size | 0.131 | −0.371 | −0.088 | −0.007 | 0.024 | 1.000 | | | | | |
| (7) CEO duality | 0.061 | 0.347 | −0.175 | 0.106 | 0.193 | 0.182 | 1.000 | | | | |
| (8) Non-executive director | 0.103 | −0.319 | 0.245 | −0.056 | 0.015 | 0.139 | −0.108 | 1.000 | | | |
| (9) Female representation | −0.252 | 0.254 | 0.013 | −0.131 | 0.159 | −0.283 | 0.036 | −0.038 | 1.000 | | |
| (10) Board ownership | 0.037 | 0.166 | −0.028 | −0.009 | 0.092 | −0.159 | 0.340 | −0.159 | 0.115 | 1.000 | |
| (11) Board compensation | 0.052 | −0.165 | 0.032 | 0.333 | 0.152 | 0.516 | 0.236 | 0.087 | −0.087 | 0.003 | 1.000 |

**Table 7.** Matrix of correlations (Ghana).

| Variables | (1) | (2) | (3) | (4) | (5) | (6) | (7) | (8) | (9) | (10) | (11) |
|---|---|---|---|---|---|---|---|---|---|---|---|
| (1) Return on invested capital | 1.000 | | | | | | | | | | |
| (2) Return on net operating assets | 0.523 | 1.000 | | | | | | | | | |
| (3) Tobin's Q | 0.014 | 0.041 | 1.000 | | | | | | | | |
| (4) Board tenure | −0.076 | −0.227 | −0.003 | 1.000 | | | | | | | |
| (5) Number of board meetings | −0.335 | −0.309 | −0.035 | 0.375 | 1.000 | | | | | | |
| (6) Board size | −0.136 | −0.176 | −0.180 | 0.168 | 0.418 | 1.000 | | | | | |
| (7) CEO duality | 0.041 | 0.035 | 0.259 | −0.164 | −0.173 | −0.087 | 1.000 | | | | |
| (8) Non-executive director | 0.054 | −0.026 | −0.116 | 0.100 | 0.182 | 0.289 | 0.044 | 1.000 | | | |
| (9) Female representation | 0.011 | 0.151 | 0.268 | −0.275 | −0.372 | −0.348 | 0.454 | −0.408 | 1.000 | | |
| (10) Board ownership | −0.270 | −0.078 | −0.067 | −0.047 | −0.028 | −0.276 | −0.146 | −0.048 | −0.004 | 1.000 | |
| (11) Board compensation | 0.061 | −0.014 | −0.089 | 0.061 | 0.183 | 0.162 | −0.082 | 0.049 | −0.335 | −0.072 | 1.000 |

### 4.3. Variance Inflation Analysis

Table 8 shows the results for the variance inflation factor (VIF). The VIF is used to detect the presence of multicollinearity among the independent variables. If the independent variables, which are corporate governance variables, are highly correlated with each other, then there is a problem of multicollinearity. When the VIF is greater than 10 [103], then there is a problem of multicollinearity. The results from Table 9 show that there is no multicollinearity since the highest VIF among the three countries' corporate governance variables is 1.363 and the mean VIF is 1.406.

**Table 8.** Variance inflation factor.

| | South Africa | | Nigeria | | Ghana | |
|---|---|---|---|---|---|---|
| | VIF | 1/VIF | VIF | 1/VIF | VIF | 1/VIF |
| Non-executive director | 1.166 | 0.858 | 1.057 | 0.946 | 1.363 | 0.734 |
| Board compensation | 1.151 | 0.869 | 1.712 | 0.584 | 1.171 | 0.854 |
| Number of board meetings | 1.134 | 0.882 | 1.086 | 0.921 | 1.449 | 0.69 |
| Board size | 1.128 | 0.887 | 1.7 | 0.588 | 1.447 | 0.691 |
| Board tenure | 1.123 | 0.891 | 1.242 | 0.805 | 1.205 | 0.83 |
| Female representation | 1.09 | 0.917 | 1.164 | 0.859 | 2.061 | 0.485 |
| CEO duality | 1.051 | 0.951 | 1.286 | 0.778 | 1.42 | 0.704 |
| Board ownership | 1.048 | 0.954 | 1.223 | 0.818 | 1.134 | 0.882 |
| Mean VIF | 1.111 | | 1.309 | | 1.406 | |

**Table 9.** The effect of corporate governance on financial performance (GMM results).

| Variable | South Africa | Nigeria | Ghana | South Africa | Nigeria | Ghana | South Africa | Nigeria | Ghana |
|---|---|---|---|---|---|---|---|---|---|
| | | Model 1 | | | Model 2 | | | Model 3 | |
| LAG | 0.804 ** | −0.424 ** | −0.274 ** | −0.107 ** | 0.316 | 0.182 ** | −0.233 | 0.039 | 0.101 |
| | (0.357) | (0.214) | (0.122) | (0.053) | (0.22) | (0.092) | (0.547) | (0.05) | (0.121) |
| Board tenure | 0.518 *** | 0.795 | 0.385 * | −0.899 *** | −0.489 ** | 0.251 | 0.006 ** | 0.014 ** | 0.042 ** |
| | (0.183) | (2.709) | (0.197) | (0.218) | (0.226) | (0.280) | (0.003) | (0.007) | (0.021) |
| Number of board meetings | −0.698 | −0.743 | −0.230 * | 0.197 ** | 0.839 ** | 0.344 *** | 0.013 *** | −0.050 *** | 0.105 *** |
| | (0.519) | (1.986) | (0.118) | (0.097) | (0.403) | (0.107) | (0.004) | (0.002) | (0.003) |
| Board size | −0.865 ** | 0.295 | 0.252 | −0.405 | −0.401 *** | −0.139 | −0.001 | −0.007 | −0.049 |
| | (0.439) | (1.27) | (0.159) | (0.587) | (0.125) | (0.536) | (0.002) | (0.005) | (0.052) |
| CEO duality | −0.933 | 0.111 | 0.430 ** | 0.595 | 0.700 | −0.925 | 0.014 | −0.025 * | 0.352 |
| | (0.589) | (0.418) | (0.210) | (0.406) | (0.991) | (2.237) | (0.015) | (0.013) | (0.362) |
| Non-executive director | 0.329 *** | 0.015 | −0.229 * | 0.158 ** | 0.613 * | −0.742 | 0.011 *** | 0.192 *** | 0.004 |
| | (0.111) | (0.129) | (0.126) | (0.074) | (0.32) | (0.725) | (0.001) | (0.048) | (0.003) |
| Female representation | 0.05 | −0.171 | −0.842 ** | 0.454 *** | 0.137 *** | 0.151 *** | 0.092 *** | 0.035 *** | 0.011 * |
| | (0.087) | (0.314) | (0.415) | (0.106) | (0.056) | (0.033) | (0.025) | (0.011) | (0.006) |
| Board ownership | −0.253 ** | 0.138 ** | 0.412 * | 0.14 | −0.230 | −0.629 | 0.003 | 0.015 *** | 0.017 *** |
| | (0.100) | (0.056) | (0.226) | (5.602) | (0.170) | (1.662) | (0.013) | (0.003) | (0.006) |
| Board compensation | 0.108 | −0.969 | −0.920 | 0.267 *** | 0.115 ** | 0.799 *** | 0.075 *** | 0.017 *** | 0.029 *** |
| | (0.083) | (3.778) | (0.884) | (0.115) | (0.005) | (0.261) | (0.009) | (0.001) | (0.002) |
| Number of observations | 270 | 150 | 100 | 270 | 150 | 100 | 270 | 150 | 100 |
| Country dummies | Yes | Yes | Yes | Yes | Yes | Yes | Yes | Yes | Yes |
| Years dummies | Yes | Yes | Yes | Yes | Yes | Yes | Yes | Yes | Yes |
| AR (1) | 0.000 | 0.00 | 0.000 | 0.000 | 0.001 | 0.000 | 0.000 | 0.001 | 0.000 |
| AR (2) | 0.637 | 0.472 | 0.642 | 0.567 | 0.984 | 0.613 | 0.453 | 0.341 | 0.123 |
| Sargan | 0.346 | 0.623 | 0.665 | 0.223 | 0.633 | 0.473 | 0.352 | 0.435 | 0.224 |

*** $p < 0.01$, ** $p < 0.05$, * $p < 0.1$.

*4.4. Results for GMM Estimation*

Table 9 shows the results obtained for the GMM. The statistically insignificant results for the Sargan tests show the validity or fitness of the model [104]. The Sargan test results for all the models showed statistically insignificant results, indicating the validity and fitness of the model. The statistically significant results for AR (1) and the statistically insignificant results for AR (2) also confirm the fitness of the model, whereas the statistically insignificant results for AR (2) also indicated that there was no autocorrelation [99].

Model 1: The effect of corporate governance on return on invested capital

Table 9 shows the effect of corporate governance on ROIC for the three manufacturing firms in Africa. The lags of the dependent variables in the regression model, which is a dynamic panel, have a statistically positive effect on the ROIC of South African manufacturing firms. The dynamic panel has a negative and statistically significant effect on the ROIC of the manufacturing firms in Nigeria and Ghana. Board tenure was found to have a positive and statistically significant effect on the ROIC of the manufacturing firms in South Africa and Ghana, while board tenure had a positive but statistically insignificant effect on the ROIC of the manufacturing firms in Nigeria. The more years board members

spend on the corporate board, the better the financial performance, which is one of the main benefits of adopting suitable corporate governance mechanisms and structures [105]. The positive effect of board tenure on ROIC resulted from directors with more experience and familiarity with the company's operations who make recommendations [62].

The number of board meetings in a year affects ROIC in all three countries. The negative effect of board meetings on ROIC may be because infrequent board meetings did not give board members enough time to address pressing concerns [69]. However, the number of board meetings by the board members in Ghana had a statistically significant effect on ROIC, while statistically insignificant effects were found among the firms in South Africa and Nigeria. The board size of the manufacturing firm in South Africa had a negative and statistically significant effect on ROIC, while in Ghana and Nigeria, the board size had a positive effect on ROIC. The positive effect of board size on ROIC may be that corporations with a bigger board size produce better financial reporting due to the board's increased inspection and monitoring function [71]. The negative effect of board size on ROIC may be that when firms have many directors, it is difficult for them to monitor the management [73]. The study showed a negative effect of CEO duality on the ROIC of the firms in South Africa. In Nigeria and Ghana, the study showed a positive effect of CEO duality on ROIC. However, CEO duality was found to have a statistically significant effect on the ROIC of the firms in Ghana. The study showed a positive and statistically significant effect of non-executive directors on the ROIC of the firms in South Africa, while a negative and statistically significant effect was found among the firms in Ghana. It has been stated [106] that companies with more independent directors have a higher return on investment because they can better supervise their operations. In Nigeria, the positive effect of non-executive directors on ROIC was found.

The study showed a positive effect of female representation on board on ROIC, while a negative effect was found among the firms in Nigeria and Ghana. The positive effect of female representation on the board confirms stakeholder theory, which says that having more women on the board means paying more attention to and focusing on the demands and interests of stakeholders, which improves board performance and boosts company success [84]. However, female presentation on boards has a statistically significant effect on the ROIC of the firms in Ghana. In South Africa, the study showed that board shareholding had a negative and statistically significant effect on ROIC, while in Nigeria and Ghana, it had a positive and statistically significant effect. The study showed a positive effect of board members' compensation on the ROIC of the firms in South Africa, while the firms in Nigeria and Ghana found a negative effect. The positive effect of board compensation packages is in line with the agency theorists, who say that the most effective means of reducing agency difficulties are for the board of directors to keep a close eye on management and offer board members financial incentives [96].

Model 2: The Effect of Corporate Governance on RONOA

Table 9 shows the effect of corporate governance on RONOA. The dynamic panel, which was the lag of the dependent variable included in the regression model, had a negative and statistically significant effect on the RONOA of the firms in South Africa, while a positive effect was found among the firms in Nigeria and Ghana. However, the dynamic panel was found to have a statistically significant effect on the RONOA of the firms in Ghana. The study showed a negative effect of board tenure on the RONOA of the firms in South Africa and Nigeria, but the firm in Ghana found a positive effect. The tenure of the board members in South Africa and Nigeria had a statistically significant effect on RONOA.

The study showed a positive and statistically significant effect of the number of board meetings on the RONOA of all the firms in the three countries. The result is aligned with the findings of [66], who found that the corporate governance index has a positive and statistically significant effect on return on net worth. The board size was found to have a detrimental effect on the RONOA of all firms in the three countries. However, the board size was found to have a statistically significant effect on the RONOA of the firms in Nigeria.

The study showed that in South Africa and Ghana, CEO duality had a positive effect on RONOA, but the firms in Ghana found a negative effect. The study showed a positive and statistically significant effect of female representation on RONOA of all the South African, Nigerian, and Ghanaian firms. The authors in [107] found that having a female CEO or female director on the board of directors will result in better financial performance, consistent with the study results.

The study showed that non-executive directors had a positive and statistically significant effect on the RONOA of firms in South Africa and Nigeria but a statistically insignificant effect on firms in Ghana. Research by Yasser [108] found that non-executive directors had a detrimental effect on the cash flow from operating activities, which contradicts the study outcome. The shareholdings of the board members were found to have a positive effect on the RONOA of the firms in South Africa, while a detrimental effect was found among the firms in Nigeria and Ghana. Gaur, Bathula, and Singh [109] found that lack of ownership concentration had a detrimental effect on financial performance, and the results are compatible with the results obtained for Nigerian and Ghanaian firms. The compensation to the board members was found to have a positive and statistically significant effect on the RONOA of the South African, Nigerian, and Ghanaian firms. The findings contradict the results of Jiang and Zhang [110], who found that compensation paid to board members has a negative and statistically significant impact on ROA.

Model 3: The effect of corporate governance on Tobin's Q

Table 9 shows the effect of the corporate governance of the manufacturing firms in South Africa, Nigeria, and Ghana on Tobin's Q. The dynamic panel was found to have a deleterious effect on Tobin's Q for the firms in South Africa, while a positive effect was found for the firms in Nigeria and Ghana. The study showed a positive and statistically significant effect of board tenure on Tobin's Q of the manufacturing firms in South Africa, Nigeria, and Ghana. In South Africa and Ghana, the study showed that the number of board meetings has a positive and statistically significant effect on Tobin's Q, while a negative and statistically significant effect was found for firms in Nigeria. Research by Yasser [108] found that board meetings have a positive and statistically significant effect on Tobin's Q, which is consistent with the findings for the firms in South Africa and Ghana but contradicts those of Nigeria.

The size of the boards of the manufacturing firms in South Africa, Nigeria, and Ghana had a deleterious effect on Tobin's Q. The results contradict the findings by [109–111], who find a positive effect of board size on Tobin's Q. The study showed a positive effect of CEO duality on Tobin's Q of the firms in South Africa and Ghana but a negative and statistically significant effect on Tobin's Q of the firms in Nigeria. The results obtained for the firms in Nigeria are consistent with [112] 's findings. The findings by [113] contradict the study results; they found a negative and statistically significant effect on Tobin's Q. The study showed that non-executive directors had a positive effect on Tobin's Q of the firms in South Africa, Nigeria, and Ghana. The non-executive director was found to have a statistically significant effect on Tobin's Q of the firms in South Africa and Nigeria. A study conducted in Ghana found that non-executive board members had a positive and statistically significant effect on Tobin's Q, which is commensurate with the study results for the firms in Ghana [114]. Akbar [115] found that the non-executive board has a negative and statistically significant effect on Tobin's, opposite to the study findings.

The study showed a positive and significant effect of female representation on Tobin's Q for all the South African, Nigerian, and Ghanaian firms. The study results are in line with the findings of [115]. A firm's performance is enhanced and assured if there are corporate governance mechanisms to minimize agency conflict by including females on the board [116]. The study showed a positive effect of board members' shareholdings on Tobin's Q for all the South African, Nigerian, and Ghanaian firms. However, board members' shareholdings in the firms in Nigeria and South Africa had a statistically significant effect on Tobin's Q, consistent with the findings obtained by [117]. The compensation paid to the board members of the firms in South Africa, Nigeria, and Ghana has a positive and

statistically significant effect on Tobin's Q. The findings are incongruous with the results obtained by [76,118], who find negative and statistically insignificant CEO compensation on Tobin's Q.

### 4.5. Results for FMOLS Estimation

Table 10 shows the FMOL results of corporate governance on the financial performance of manufacturing firms in South Africa, Nigeria, and Ghana.

**Table 10.** The effect of corporate governance on financial performance (FMOLS).

| Variable | South Africa | Nigeria | Ghana | South Africa | Nigeria | Ghana | South Africa | Nigeria | Ghana |
|---|---|---|---|---|---|---|---|---|---|
| | | Model 1 | | | Model 2 | | | Model 3 | |
| Board tenure | 0.562 *** | 0.011 | 0.226 | −0.133 ** | −0.421 ** | 0.312 | 0.005 *** | 0.019 ** | 0.015 |
| | (0.213) | (0.826) | (0.698) | (0.057) | (0.205) | (0.203) | (0.001) | (0.009) | (0.023) |
| Number of board meetings | −0.114 *** | −0.099 | −0.578 *** | 0.1455 ** | 0.541 ** | 0.997 ** | 0.004 *** | −0.045 *** | 0.057 *** |
| | (0.026) | (0.761) | (0.142) | (0.068) | (0.268) | (0.414) | (0.001) | (0.009) | (0.006) |
| Board size | −0.18 | 0.268 | 0.992 * | −0.118 ** | −0.430 *** | −0.188 | −0.002 | −0.004 * | −0.035 * |
| | (0.205) | (0.409) | (0.576) | (0.055) | (0.136) | (0.168) | (0.001) | (0.002) | (0.019) |
| CEO duality | −0.351 | 0.114 | 0.137 | 0.029 | 0.220 *** | −0.913 | 0.022 * | −0.024 ** | 0.203 ** |
| | (2.287) | (0.196) | (0.308) | (6.107) | (0.065) | (0.896) | (0.013) | (0.012) | (0.100) |
| Non-executive director | 0.077 | 0.109 | −0.059 | 0.197 | 0.423 | 0.134 | 0.098 *** | 0.002 *** | 0.001 |
| | (0.051) | (0.079) | (0.045) | (0.137) | (0.264) | (0.132) | (0.029) | (0.004 | (0.001) |
| Female representation | 0.084 * | −0.221 *** | −0.063 *** | 0.584 *** | 0.138 *** | 0.213 ** | 0.079 *** | 0.065 *** | 0.004 |
| | (0.048) | (0.071) | (0.017) | (0.129) | (0.038) | (0.103) | (0.002) | (0.025) | (0.003) |
| Board ownership | −0.107 * | 0.299 | 0.353 *** | −0.127 | −0.146 | −0.381 | 0.006 * | 0.025 *** | 0.013 *** |
| | (0.058) | (0.324) | (0.088) | (0.154) | (0.108) | (0.257) | (0.003) | (0.002) | (0.003) |
| Board compensation | 0.115 | −0.284 | −0.112 | 0.708 ** | 0.847 ** | 0.208 *** | 0.011 *** | 0.019 *** | 0.028 *** |
| | (0.109) | (1.137) | (0.096) | (0.291) | (0.419) | (0.080) | (0.003) | (0.004) | (0.031) |
| Constant | 7.964 | 3.307 | 28.699 *** | 0.719 ** | 1.054 *** | 5.72 ** | −0.092 ** | −0.097 * | 0.114 |
| | (7.161) | (9.251) | (9.686) | (0.191) | (0.309) | (2.818) | (0.04) | (0.055) | (0.316) |
| Number of observations | 290 | 170 | 140 | 290 | 170 | 140 | 290 | 170 | 140 |
| Country dummies | Yes | Yes | Yes | Yes | Yes | Yes | Yes | Yes | Yes |
| Years dummies | Yes | Yes | Yes | Yes | Yes | Yes | Yes | Yes | Yes |
| R-square | 0.334 | 0.438 | 0.123 | 0.297 | 0.573 | 0.441 | 0.621 | 0.452 | 0.301 |

*** $p < 0.01$, ** $p < 0.05$, * $p < 0.1$.

Model 1: The effect of corporate governance on return on invested capital (ROIC)

The result indicates that board tenure had a positive and statistically significant effect on the ROIC of the firms in South Africa, while a statistically insignificant effect was found for the firms in Nigeria and South Africa. The number of board meetings in a year had a negative and statistically significant effect on the ROIC of the firms in South Africa and Ghana, while it has a negative and statistically insignificant effect on the ROIC of the firm in Nigeria. The board size was found to have a positive and statistically significant effect on the ROIC of the firms in Ghana, and a statistically insignificant effect of board size on ROIC was found for the firms in South Africa and Nigeria. CEO duality and non-executive directors were found to have a statistically insignificant effect on ROIC for all the firms in South Africa, Nigeria, and Ghana. In South Africa, female representation on boards was found to have a positive and statistically significant effect on ROIC, while a negative and statistically significant effect of female representation on ROIC was found for the firms in Nigeria and Ghana. The shareholdings of the board members in the manufacturing firm were found to have a negative and statistically significant effect on the ROIC of the firms in South Africa and a positive and statistically significant effect of the shareholdings of the board members on the ROIC of the firms in Ghana. The study showed a positive and statistically insignificant effect of board members' shareholdings on ROIC for the firms in Nigeria. The results show that board compensation had a statistically insignificant effect on ROIC for all the South African, Nigerian, and Ghanaian firms. The findings are consistent with the main results presented in Table 9.

Model 2: The effect of corporate governance on RONOA

The study showed a negative and statistically significant effect of board tenure on the RONOA of the firms in South Africa and Nigeria but a positive and statistically insignificant effect of board tenure on the RONOA of the firms in Ghana. In all three countries, South Africa, Nigeria, and Ghana, the study showed a positive and statistically significant effect of board meetings on RONOA. The board size was found to have a negative and statistically significant effect on RONOA for the firms in South Africa and Nigeria, but a statistically insignificant effect of the board size on RONOA was found for the firms in Ghana. CEO duality was found to have a positive and statistically significant effect on RONOA for the firms in Nigeria, but a statistically insignificant effect on RONOA was found for the firms in South Africa and Ghana. Non-executive directors were found to have a statistically insignificant effect on RONOA for the manufacturing firms in South Africa, Nigeria, and Ghana. Female representation on the board and compensation paid to the board members were found to have a positive and statistically significant effect on RONOA for all the firms in South Africa, Nigeria, and Ghana. The shareholding by the board members was found to have a negative and statistically insignificant effect on RONOA for all the manufacturing firms in South Africa, Nigeria, and Ghana. The finding obtained is consistent with the main results presented in Table 9.

Model 3: The effect of corporate governance on Tobin's Q

The study showed a positive and statistically significant effect of board tenure on Tobin's Q for the firms in South Africa and Nigeria but a statistically insignificant effect for the firms in Ghana. The number of board members' meetings in a year was found to have a positive and statistically significant effect on Tobin's Q for the firms in South Africa and Ghana but a negative and statistically significant effect for the firm in Nigeria. Research by Arora and Sharma [119] found that board meetings have a positive and statistically significant effect on Tobin's Q, which is in accordance with the results obtained for the firms in South Africa and Ghana but contradicts that of the firms in Nigeria.

The results show that the board size of the firms in Nigeria and Ghana had a negative and statistically significant effect on Tobin's Q but a negative and statistically insignificant effect on the firms in South Africa. The findings for both countries contradict the results discovered by [119], who found a positive and statistically significant effect of board size on Tobin's Q. The study showed a positive and statistically significant effect of CEO duality on Tobin's Q for the firms in South Africa and Ghana but a negative and statistically significant effect for the firms in Nigeria. The positive and statistically significant effect of CEO duality on Tobin's Q is akin to the findings obtained by [120].

The results show that the non-executive director and female representation on the board had a positive and statistically significant effect on Tobin's Q for the firms in South Africa and Nigeria but a statistically insignificant effect for the firms in Ghana. The positive effect of female representation on boards agrees with the findings [121]. A negative and statistically significant female representation on board was discovered by [122], which is a confutation of the study outcome. The shareholdings by the board members and compensation paid to the board members were found to have a positive and statistically significant effect on Tobin's Q for all the firms in Ghana, Nigeria, and South Africa. Research conducted in Pakistan found that the corporate governance index, which includes board member compensation, has a negative effect on Tobin's Q, which shows inconsistency with the study results [118]. The finding obtained is consistent with the main results presented in Table 9.

## 5. Conclusions

The effect of corporate governance on financial performance has been investigated to determine how corporate governance mechanisms affect financial performance. The study compares the effects of the corporate governance structures of the manufacturing firms in Nigeria, Ghana, and South Africa on ROIC, RONOA, and Tobin's Q. The study results contribute to building investors' confidence in the governance structures of manufacturing firms in Africa.

The study showed that board tenure had a positive and statistically significant effect on the ROIC of the firms in South Africa and Ghana, but a statistically insignificant effect was found for the firms in Nigeria. The shareholdings by board members were found to have a negative and statistically significant effect on ROIC for the firms in South Africa and a positive and statistically significant effect for the firms in Nigeria and Ghana. The meetings by the board members were found to have a positive and statistically significant effect on RONOA for all of the firms in South Africa, Ghana, and Nigeria. Board meetings were found to have a positive and statistically significant effect on Tobin's Q of the manufacturing firms in South Africa and Ghana but a negative and statistically significant effect on the firms in Nigeria, which is consistent with previous studies [119]. The study showed that female representation on the board and compensation paid to board members have a positive and statistically significant effect on RONOA and Tobin's Q for all the firms in South Africa, Nigeria, and Ghana. The finding is consistent with previous studies [123].

## 6. Implications for Managerial Decisions

The three independent variables used, ROIC, RONOA, and Tobin's Q, give a true reflection of, or measure, the actual performance of the manufacturing firms. ROIC is used to assess the efficiency of manufacturing firms in allocating shareholder money to a profitable investment. RONOA measures manufacturing companies' profit by using only their operating assets, excluding investments. RONOA indicates whether the management of the corporation and the board of directors are using the corporation's operating assets effectively. Tobin's Q measures the corporation's market value in relation to its assets. Market values measure the shareholders' wealth, one of the shareholders' objectives. The corporate governance structure is pivotal in building investors' confidence and encouraging them to continue investing in the corporation. The results show that the corporate structure of the firms in South Africa, Nigeria, and Ghana has a significant effect on the financial performance of the manufacturing companies. When board members are compensated for their work, it improves the corporation's financial performance. Compensation gives them extrinsic motivation to monitor, influence, and control the activities of the management well and act in the best interest of the shareholders.

In addition, females on boards also contribute significantly to the financial performance of the firms. The number of years that the board members spent on the board gives them experience when it comes to approving and monitoring the activities of the business. The longer they serve on the board, the more they become experienced and contribute to financial performance. The meetings of board members also contribute significantly to the financial performance of the manufacturing companies. During board meetings, any activities that would contribute negatively to the financial performance of the firms are discussed and dealt with. In corporate governance, where board members frequently meet to discuss the activities of the corporation, the inclusion of females on the board, the firm having a larger board size, and the board members owning shares in the corporation are all suitable corporate governance mechanisms to contribute to the better financial performance of the manufacturing firms in South Africa, Nigeria, and Ghana.

## 7. Limitations of the Study

The first limitation of the study was that most of the firms listed on the Ghanaian and Nigerian stock markets did not have their data between 2011 and 2020, which limited the number of firms that were included. Moreover, some manufacturing companies' accounts were not audited but had published their annual report. These companies were not included in the study. The data obtained for the manufacturing companies covered more than 10 years; from 2005 to 2020, the manufacturing firms in Ghana and Nigeria did not have their data up to that period, which limited the years that the data were used for the study. The second limitation of the study was that it focused only on South Africa, Nigeria, and Ghana, but these are not the only countries in Africa that implement corporate governance. Comparing corporate governance between five and ten countries will give a

better indication of how corporate governance influences financial performance. However, most of the companies in the African countries did not have their data in the Thomson Reuters DataStream. Their data were supposed to be extracted from their annual report. However, most companies do not publish their annual report. This made it difficult to cover them in the study.

## 8. Direction for Future Studies

Future studies should compare the effect of corporate governance on financial performance between Africa, America, Asia, and Europe to indicate better how corporate governance in those continents contributes to financial performance. Moreover, future studies should include corporate governance variables such as board ethnicities, board members' affiliations, and staggered board structure to find out their effect on financial performance. Further, similar studies that are conducted should compare the governance mechanisms and their effects on performance between West, South, Central, North, and East African countries. Future studies can also include the effect of mediating variables such as earning management, working capital, financial innovation, shareholder confidence, and business growth on the relationship between financial performance and corporate governance. Future studies should also consider adding institutional holdings, foreign holdings, and the age of the board of directors to ascertain their effect on financial performance.

**Author Contributions:** The data for the study were downloaded and extracted by L.M.; The analysis was performed by M.A.B. and L.M.; Wrote the introduction and literature review of the paper. The analysis and discussion were done by M.A.B. and M.A.B.; The supervisor, ultimately approved the work. Both authors have consented to publish this paper. All authors have read and agreed to the published version of the manuscript.

**Funding:** This research received no external funding.

**Institutional Review Board Statement:** Not applicable.

**Informed Consent Statement:** Not applicable.

**Data Availability Statement:** The related data sets are available from the corresponding author upon request.

**Conflicts of Interest:** The authors declare no conflict of interest.

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
