# Peer review of "Sound Corporate Governance and Financial Performance: Is There a Link? Evidence from Manufacturing Companies in South Africa, Nigeria, and Ghana"

_sustainability, doi:10.3390/su15129263_

Round 1
Reviewer 1 Report
The issues raised in the manuscript “Sound corporate governance and financial performance: Is there a link? Evidence from manufacturing companies in South Africa, Nigeria, and Ghana” are important, actual, and deserve in-depth research. However, in my opinion, the manuscript still needs a little bit work.
My suggestions:
· It would be clearer for the reader if the hypotheses were placed in the table. Results of studies, i. e. information which hypothesis is confirmed, can also be presented in tabular or graphical form.
· The literature review on existing studies on the relationship between corporate governance and financial performance should be deepened. When presenting theories relevant to the corporate sustainability, it is not clearly indicated how they relate to the research carried out.
· There is too much text describing the content of the tables (for example in 4.1 Section). Reader can see what is in a table. It is not necessary to write it once again.
· Instead, it would be better to put the results in some context, rather than just describing the contents of the table.
· Last paragraph in the Introduction section should present the structure of the manuscript.
· In conclusion section it is not necessary to repeat the methodology in such a detail.
The English language is understandable, the manuscript needs only proofreading.
Author Response
My suggestions:
- It would be clearer for the reader if the hypotheses were placed in the table. Results of studies, i. e. information which hypothesis is confirmed, can also be presented in tabular or graphical form.
Response
We are so grateful to you for your time reading the manuscript and your contributions to shaping the paper. We have put all the hypotheses in Table 1. The only thing we could not put on a table was the hypothesis that was confirmed and accepted. We have three dependent variables and three countries, and we have to do it for each country. That would give us more tables. One of the reviewers suggested reducing the number of tables presented in the work. However, if there is a way that you want us to do it, please teach us to follow it. We love your comments.
Table 1: Summary of hypothesis
Numbers |
Hypothesis |
Hypothesis (H1) |
Board tenure statistically significantly affects manufacturing firms' financial performance. |
Hypothesis (H2) |
The number of board meetings statistically significantly affects the manufacturing firm's financial performance. |
Hypothesis (H3) |
Board size statistically significantly affects the manufacturing firm's financial performance. |
Hypothesis (H4) |
The CEO duality statistically significantly affects the manufacturing firm's financial performance. |
Hypothesis (H5) |
Non-executive directors have a statistically significant effect on the manufacturing firm's financial performance |
Hypothesis (H6) |
Female presentation on the corporate board statistically significantly affects the manufacturing firm's financial performance. |
Hypothesis (H7) |
Board ownership statistically significantly affects the manufacturing firm's financial performance. |
Hypothesis (H8) |
Board compensation has a statistically significant effect on the manufacturing firm's financial performance. |
- The literature review on existing studies on the relationship between corporate governance and financial performance should be deepened. When presenting theories relevant to the corporate sustainability, it is not clearly indicated how they relate to the research carried out.
Responses
We thank you a lot for the insight. We have added more empirical literature to the study. We have expanded the theories to relate them to the study carried out. We are grateful for such insight and appreciate it so much.
Dzingai & Fakoya [19] used companies in the mining sector listed between 2010 and 2015 and found that board size has a negative and statistically insignificant effect on ROE, while board independence has a positive and statistically significant effect on ROE. Toshiba et al. [20] found that corporate attributes had a positive and statistically significant effect on financial performance on manufacturing companies during and after the 2008 financial crisis. Karaye & Büyükkara [21] used companies listed in Johannesburg stock market data between 2013 and 2019 and found that the corporate governance index has a positive and statistically significant relationship with ROA. In Nigeria, Urhoghide & Omolaye [22] used 14 oil and gas companies listed in the Nigerian Stock Market between 2008 and 2015. They found that board size and female presentation on the board have positive and statistically significant effects on financial performance. Yusuf, Bambale, and Abdullahi [23] used 8 financial institutions listed in the Nigerian stock market in 2015 and found that board size and composition have positive and statistically insignificant effects on financial performance. In Ghana, Musah & Adutwumwaa [24] used 10 rural banks' years of data between 2010 and 2019 and found that board size and CEO duality have positive and statistically insignificant effects on ROA and ROE. Boachie [25] used 23 financial institutions’ data sets between 2006 and 2018, and the study showed that CEO duality and non-executive directors have a positive and statistically significant relationship with financial performance. Aside from South Africa, Nigeria, and Ghana, similar studies have been conducted across the globe. Hakimah et al. [26] used 50 samples of Indonesian SMEs between 2013 and 2017 and found that board size and female presentation have a positive and statistically significant effect on ROA. Kyere & Ausloos [27] used 252 firms listed on the London Stock Exchange in the year 2014 and found that board size, independent directors, and CEO duality have a positive and statistically significant relationship with ROA, while a positive and statistically insignificant relationship was found on Tobin's Q. Al-Homaidi [28] used 30 banks between 2013 and 2016 listed on the Bombay Stock Exchange and found a statistically significant relationship with ROA
Agency theory
Managerial decisions in today's widely owned corporations deviate from what is necessary to maximize shareholder returns. According to agency theory, managers are agents, and owners are principals; an agency loss occurs when owners' returns on their residual claims are lower than if the owners themselves exercised direct management of the firm [32]. Mechanisms for minimizing agency losses are outlined in the agency theory. They include incentive programs for managers that provide financial rewards for maximizing shareholder value. Such strategies usually involve top executives buying shares, sometimes at a discount, to align their financial interests with shareholders [33]. Several similar systems relate CEO salary and benefits to shareholder returns and defer a portion of executive income to the future to promote long-term value maximization and discourage short-term executive activity that undermines business value [34]. Agency theory could affect the financial performance of manufacturing companies. When the managers of an organization have different goals from the shareholders and the management withholds essential information about manufacturing companies from the shareholders, this may impact the business's financial performance. When the shareholders of the manufacturing companies would not be willing to contribute addi-tional funds to the corporation when they found that their objectives and information received differed from those of the managers.
Stewardship theory
The stewardship theory of corporate governance downplays the possibility of tensions between business management and shareholders and favours an insider-dominated board of directors [35]. The idea presupposes that manager are trust-worthy and value their reputations highly. As a result, the market for highly regarded managers, who command higher salaries, is the essential tool to regulate behavior [36]. Financial reporting, disclosure, and auditing are still essential, but there is a fundamental assumption that these practices are necessary to confirm management's inherent integrity. One of the ways the managers account to the corporation's shareholders is through annual general meetings. When the managers did not account to the share-holders how effectively they used the corporation's assets to generate enough earnings, represented by ROA and ROE. The shareholders, through the board of directors, may not approve any assets the managers may need in the future to run the corporation, which could affect the corporation's financial performance.
Stakeholder Theory
According to stakeholder theory, a company or organization must serve multiple constituencies. According to stakeholder theory, everyone who impacts the company or its operations in any way is a stakeholder, including environmental organizations, clients, employees, local communities, suppliers, governmental organizations, and more [37]. According to stakeholder theory, organizations and businesses will be more successful in the long run if they prioritize their constituents' needs. According to shareholder theory, a company's primary purpose should be to advance the interests of its shareholders. Shareholder theory translates to making more profit at any cost business philosophy because shareholders are most concerned with financial growth [38]. The success of the corporation depends on its primary and secondary stakeholders. The corporation's failure to meet or address its stakeholders' needs may lead to its collapse in the future. For example, suppose the manufacturing companies fail to honor their tax obligations to the government. In that case, it may affect their operations due to the frustration they would face from the government, or the government could collapse the corporation. Therefore, stakeholder theory is vital to a manufacturing company's corporate governance and financial performance.
Resource Dependency Theory
According to resource dependency theory [39], the board of directors plays a pivotal role in securing the necessary external resources for an organization. Theorists of resource dependency emphasize the importance of having third-party representatives oversee the acquisition of essential resources. The availability of resources improves organizational effectiveness, productivity, and survival. The directors give the company credibility [40] and provide information, expertise, and connections to important stakeholders like government policymakers, consumers, and business partners. The smooth running of a corporation depends on its resource. Failure to supply the corporation with resources in terms of processing inputs could affect the corporation's performance. Every successful corporation depends on its inputs, such as human and capital resources.
- There is too much text describing the content of the tables (for example in 4.1 Section). Reader can see what is in a table. It is not necessary to write it once again. Instead, it would be better to put the results in some context, rather than just describing the contents of the table.
Responses
We have restructured it according to your unique suggestions. This was the original one submitted.
Tables 2, 3, and 4 show the descriptive statistics for the three countries: South Africa, Nigeria, and Ghana. The average return on invested capital (ROIC) for the manufacturing firms in South Africa was 7.64%, that of the manufacturing firms in Nigeria was 9.427%, and that of the firms in Ghana was 1.621%. The firms in Nigeria have the highest ROIC, followed by those in South Africa and Ghana. The return on net operating assets (RONOA) for the firms in South Africa was 14.79%, Nigeria was 24.05%, and Ghana was 4.28%. Regarding RONOA, the firms in Nigeria have the highest RONOA, followed by those in South Africa and Ghana. The average Tobin's Q for the firms in South Africa was 0.11, Nigeria was 0.04, and Ghana was 0.142. The results show that all the firms in the three countries are undervalued since Tobin's Q is less than 1. Regarding Tobin's Q of the firms, the firms in Ghana have the highest, followed by those in Nigeria and South Africa.
The average board tenure for firms in South Africa is 7.852 years; in Nigeria, it is 4.67 years; and in Ghana, it is 3.9 years. Regarding the average board tenure, the board members in South Africa have the highest tenure, followed by Nigeria and Ghana. The board members in South Africa have more experience than those in Nigeria and Ghana. Board meetings are the number of times the board members meet in a year for business discussions of the firms. The average number of board meetings for the firms in South Africa was 5.34; in Nigeria, it was 5.141; and in Ghana, it was 4.61 in a year. The board members in South Africa meet more often with the firms in Nigeria and Ghana, followed by those in Nigeria. Board size is the number of board members in the corporate room. The average board size of the firms in South Africa was 11, Nigeria was 9.89, and Ghana was 8.4. The firms in South Africa have the largest boards compared to those in Nigeria and Ghana. Nigerian businesses have the second-largest boards, followed by Ghanaian businesses.
The average CEO duality for the firms in South Africa was 0.951, Nigeria was 0.482, and Ghana was 0.15. The manufacturing firms in South Africa have more dual CEO roles than those in Nigeria and Ghana. Comparison of CEO duality between Nigeria and Ghana: Nigeria has more dual CEO roles than the firms in Ghana. Non-executive directors are outside board members. The average number of non-executive directors on the boards of manufacturing firms in South Africa was 72.35%, Nigeria was 74.08%, and Ghana was 67.14%. The firms in Nigeria have a greater percentage of non-executive directors on their corporate boards than those in South Africa and Ghana. However, the firms in South Africa have a greater percentage of non-executive directors than those in Ghana. The percentage of female presentations on the board is the total number of females on the board over the total number of board members. The percentage of female directors on the boards of manufacturing firms in South Africa was 24.26%; in Nigeria, it was 17.31%; and in Ghana, it was 17.68%. The firms in South Africa have the highest number of female directors, followed by those in Nigeria and Ghana.
Board ownership is the percentage of shares the board members have in the firms. The average shareholding of the board members in South Africa was 0.562%, Nigeria was 1.119%, and Ghana was 4.97%. The board members in Ghana have a greater percentage of shares in the firms than the manufacturing firms in South Africa and Nigeria. Comparing the shareholdings of the board members between Nigeria and South Africa, the board members in Nigeria have the highest shareholdings. Outside board members are not paid but receive compensation for their activities on the firm's behalf. The average amount of board compensation to the board members of manufacturing firms in South Africa was R802,396; in Nigeria, it was ₦144,748,057; and in Ghana, it was GH¢7,358,264. The board members in Nigeria received the highest compensation, followed by the board members in Ghana, and the board members in South Africa received the lowest compensation compared to the board members in the three countries.
This is what the new one looks like, per your suggestions.
Tables 2, 3, and 4 show the descriptive statistics for the three countries: South Africa, Nigeria, and Ghana. The firms in Nigeria have the highest ROIC, followed by those in South Africa and Ghana. Regarding RONOA, the firms in Nigeria have the highest RONOA, followed by South Africa and Ghana. The results show that all the firms in the three countries are undervalued since Tobin's Q is less than 1. The firms in Ghana have the highest Tobin's Q, followed by Nigeria and South Africa. The board members in South Africa have the longest tenure, followed by Nigeria and Ghana. The board members in South Africa have more experience than those in Nigeria and Ghana. The board members in South Africa meet more often followed by Nigeria and then Ghana. Board size is the number of board members in the corporate room. The firms in South Africa have the largest number of board members compared to those in Nigeria and Ghana. Nigerian businesses have the second-largest number of board members, followed by Ghanaian businesses.
The manufacturing firms in South Africa have more dual CEO roles than those in Nigeria and Ghana. Comparison of CEO duality between Nigeria and Ghana: Nigeria has more dual CEO roles than the firms in Ghana. The firms in Nigeria have a greater percentage of non-executive directors on their corporate boards than those in South Africa and Ghana. However, the firms in South Africa have a greater percentage of non-executive directors than those in Ghana. The firms in South Africa have the highest number of female directors, followed by those in Nigeria and Ghana. The board members in Ghana have a greater percentage of shares in the firms than the manufacturing firms in South Africa and Nigeria. Comparing the shareholdings of the board members between Nigeria and South Africa, the board members in Nigeria have the highest shareholdings. Compared to the board members in the other two countries, Nigerian board members received the highest compensation. Board members in Ghana received the second-highest compensation, and South African board members received the lowest compensation
- Last paragraph in the Introduction section should present the structure of the manuscript.
Response
We have affected your recommendation.
The rest of the study is organized as follows: literature review, material and methods, empirical analysis and discussion, conclusions, managerial implications, limitations of the study, and direction for future studies.
- In conclusion section it is not necessary to repeat the methodology in such a detail.
Response
We have deleted this content from the manuscript.
One of the objectives of the business is to grow and operate in the unforeseeable future. The management of the business operations is left in the hands of management, which sometimes has different goals from the objective of the business. Failure to regulate management activities has led to the collapse of prominent businesses in Africa, Europe, and Ghana. A key example is Enron Corporation. To ensure that the objectives of the management are aligned with those of the organization and that the management acts in the best interest of the shareholders. The introduction of corporate governance has helped to reduce corporate scandals and contributed to the improvement of firms manufacturing performance. Corporate governance and financial performance have been researched for the past five years, where ROA and ROA were used as proxies for financial performance. However, ROA and ROE are not the only proxies to measure firms' financial performance. The return on invested capital (ROIC) and return on net operating assets (RONOA) have not yet been used as proxies for financial performance to find the effect of corporate governance on them.
The purpose of the study is to find the effect of corporate governance on financial performance using ROIC, RONOA, and Tobin's Q as proxies for financial performance and board tenure, the number of board meetings, the board size, CEO duality, non-executive directors, female presentation, board ownership, and board compensation as proxies for corporate governance. The study compares the effect of corporate governance on the financial performance of three African countries: South Africa, Ghana, and Nigeria. The study used 60 manufacturing companies in all, 27 from South Africa, 17 from Nigeria, and 14 from Ghana. The data used for the study for South Africa was downloaded from Thomson Reuters Eikon DataStream, and that of Ghana and Nigeria was extracted from their annual reports. The study used a purposive sampling technique to select all the manufacturing companies between 2011 and 2020. The manufacturing companies that do not have data up to the year were not included in the study. The GMM and FMOLS were the methods used to estimate the effect of corporate governance on the financial performance of manufacturing firms.
Comments on the Quality of English Language
The English language is understandable, the manuscript needs only proofreading.
Response:
We gave the manuscript to three English teachers who would do the proofreading for us. Some grammatical errors were corrected.
Submission Date
21 April 2023
Date of this review
28 Apr 2023 22:18:08

Reviewer 2 Report
1. Literature review is adequate and consistent.
2. Hypothese formutated are sound.
3. Methodology and results of the research described adequatly.
4. Discussion integrated into results presentations.
5. Conclusions are consistent.
6. Abstract should be reviewed (Nigeria and Nigeria to be coreeected).
7. References - good referencing.
Author Response
Response
We thank you a lot for the time and your effort spent to read the manuscript. We are grateful for all your inputs.
- Literature review is adequate and consistent.
- Hypotheses formulated are sound.
- Methodology and results of the research described adequately.
- Discussion integrated into results presentations.
- Conclusions are consistent.
- Abstract should be reviewed (Nigeria and Nigeria to be corrected).
Response
We thank you for the insight. We have corrected it now.
The study aimed to compare the effect of sound corporate governance on manufacturing companies in South Africa, Nigeria, and Ghana on financial performance.
- References - good referencing.
Submission Date
21 April 2023
Date of this review
01 May 2023 12:51:56

Reviewer 3 Report
The topic of the manuscript is precisely formulated.
The objective of the research is clearly explained in "Abstract": "The study aimed to compare the effect of sound corporate governance in manufacturing companies in South Africa, Nigeria, and Nigeria on financial performance". Nigeria is mentioned twice. One time it should be changed to "Ghana".
By the objective of the study, research questions in their actual version should be reduced or modified: "The study seeks to answer these questions: (1) Does corporate governance affect ROIC as a measure of the financial performance of the firms? and (2) Does corporate governance affect RONOA as a measure of the operating performance of the firms? and (3) Does corporate governance affect Tobin's Q as a measure of the market performance of the firms?"
The content and the number of references should be reduced twice. The manuscript seems like a Master's Thesis.
In "References" should be mentioned sources of data collected by authors. Only in "Conclusions" the authors wrote that "the data used for the study for South Africa was downloaded from Thomson Reuters Eikon DataStream, and that of Ghana and Nigeria was extracted from their annual reports". Such an explanation should be moved to "Sample and data".
The information in "Authors contribution" - "M.B.A, my supervisor, who finally approved the work" suggests that this manuscript is a short version or part of the thesis (Master's or Ph.D.). The original "work" description should be added to "References".
On page 10 authors wrote: “"BTE" denotes board tenure; "NBM" denotes the number of board meetings in a year; "BSZ" denotes board size; "CSY" denotes CEO duality; "NXD" denotes non-executive director; "FPN" denotes female presentation on the corporate board; "BOP" denotes board ownership; "BCN" denotes board compensation” - when 9 lines upper they wrote the same: “Board tenure – BTE, Number of board meetings – NBM […]”. Such content duplications should be reduced.
Poor English caused problems with a proper understanding of content. For example: "The study selected three countries for the study: South Africa, Nigeria, and Ghana. The three countries were chosen due to the availability of data and the fact that they are in Africa, where governance is practiced"; "Established countries like Germany, the United States, the United Kingdom, and others have taken effective measures to raise corporate governance standards and forestall future failures"; "According to a World Bank indicator for 2021, in Sub-Saharan Africa's total GDP, the manufacturing firms in Nigeria contributed a value-added value of 5%, South Africa contributed a value-added value of 15%, and Ghana contributed a value-added value of 12%"; "Ghana, Nigeria and South Africa are among the fastest-growing manufacturing industries in most African" and others.
Author Response
Reviewer three
Open Review
(x) I would not like to sign my review report
( ) I would like to sign my review report
Quality of English Language
( ) I am not qualified to assess the quality of English in this paper
( ) English very difficult to understand/incomprehensible
(X) Extensive editing of English language required
( ) Moderate editing of English language
( ) Minor editing of English language required
( ) English language fine. No issues detected
Yes |
Can be improved |
Must be improved |
Not applicable |
|
Is the content succinctly described and contextualized with respect to previous and present theoretical background and empirical research (if applicable) on the topic? |
( ) |
(x) |
( ) |
( ) |
Are all the cited references relevant to the research? |
( ) |
(x) |
( ) |
( ) |
Are the research design, questions, hypotheses and methods clearly stated? |
( ) |
(x) |
( ) |
( ) |
Are the arguments and discussion of findings coherent, balanced and compelling? |
( ) |
(x) |
( ) |
( ) |
For empirical research, are the results clearly presented? |
( ) |
(x) |
( ) |
( ) |
Is the article adequately referenced? |
( ) |
( ) |
(x) |
( ) |
Are the conclusions thoroughly supported by the results presented in the article or referenced in secondary literature? |
( ) |
(x) |
( ) |
( ) |
Comments and Suggestions for Authors
The topic of the manuscript is precisely formulated.
The objective of the research is clearly explained in "Abstract": "The study aimed to compare the effect of sound corporate governance in manufacturing companies in South Africa, Nigeria, and Nigeria on financial performance". Nigeria is mentioned twice. One time it should be changed to "Ghana".
Response
We want to thank you for spending your time and effort to read the manuscript and make suggestions that put the paper in good shape. We are grateful for your input. We have made the changes now.
The study aimed to compare the effect of sound corporate governance on manufacturing companies in South Africa, Nigeria, and Ghana on financial performance
By the objective of the study, research questions in their actual version should be reduced or modified: "The study seeks to answer these questions: (1) Does corporate governance affect ROIC as a measure of the financial performance of the firms? and (2) Does corporate governance affect RONOA as a measure of the operating performance of the firms? and (3) Does corporate governance affect Tobin's Q as a measure of the market performance of the firms?"
Response
We have done so according to your suggestion. We are grateful.
The study seeks to answer the question, "does corporate governance affect ROIC, RONOA, and Tobin's Q as a measure of the financial performance of the firms?"
The content and the number of references should be reduced twice. The manuscript seems like a Master's Thesis.
Response
We are grateful for your insight and suggestions. We have taken some content out of the manuscript. The study is not a master's thesis. It was a study that we carried out.
Tables 2, 3, and 4 show the descriptive statistics for the three countries: South Africa, Nigeria, and Ghana. The average return on invested capital (ROIC) for the manufacturing firms in South Africa was 7.64%, that of the manufacturing firms in Nigeria was 9.427%, and that of the firms in Ghana was 1.621%. The firms in Nigeria have the highest ROIC, followed by those in South Africa and Ghana. The return on net operating assets (RONOA) for the firms in South Africa was 14.79%, Nigeria was 24.05%, and Ghana was 4.28%. Regarding RONOA, the firms in Nigeria have the highest RONOA, followed by those in South Africa and Ghana. The average Tobin's Q for the firms in South Africa was 0.11, Nigeria was 0.04, and Ghana was 0.142. The results show that all the firms in the three countries are undervalued since Tobin's Q is less than 1. Regarding Tobin's Q of the firms, the firms in Ghana have the highest, followed by those in Nigeria and South Africa.
The average board tenure for firms in South Africa is 7.852 years; in Nigeria, it is 4.67 years; and in Ghana, it is 3.9 years. Regarding the average board tenure, the board members in South Africa have the highest tenure, followed by Nigeria and Ghana. The board members in South Africa have more experience than those in Nigeria and Ghana. Board meetings are the number of times the board members meet in a year for business discussions of the firms. The average number of board meetings for the firms in South Africa was 5.34; in Nigeria, it was 5.141; and in Ghana, it was 4.61 in a year. The board members in South Africa meet more often with the firms in Nigeria and Ghana, followed by those in Nigeria. Board size is the number of board members in the corporate room. The average board size of the firms in South Africa was 11, Nigeria was 9.89, and Ghana was 8.4. The firms in South Africa have the largest boards compared to those in Nigeria and Ghana. Nigerian businesses have the second-largest boards, followed by Ghanaian businesses.
The average CEO duality for the firms in South Africa was 0.951, Nigeria was 0.482, and Ghana was 0.15. The manufacturing firms in South Africa have more dual CEO roles than those in Nigeria and Ghana. Comparison of CEO duality between Nigeria and Ghana: Nigeria has more dual CEO roles than the firms in Ghana. Non-executive directors are outside board members. The average number of non-executive directors on the boards of manufacturing firms in South Africa was 72.35%, Nigeria was 74.08%, and Ghana was 67.14%. The firms in Nigeria have a greater percentage of non-executive directors on their corporate boards than those in South Africa and Ghana. However, the firms in South Africa have a greater percentage of non-executive directors than those in Ghana. The percentage of female presentations on the board is the total number of females on the board over the total number of board members. The percentage of female directors on the boards of manufacturing firms in South Africa was 24.26%; in Nigeria, it was 17.31%; and in Ghana, it was 17.68%. The firms in South Africa have the highest number of female directors, followed by those in Nigeria and Ghana.
Board ownership is the percentage of shares the board members have in the firms. The average shareholding of the board members in South Africa was 0.562%, Nigeria was 1.119%, and Ghana was 4.97%. The board members in Ghana have a greater percentage of shares in the firms than the manufacturing firms in South Africa and Nigeria. Comparing the shareholdings of the board members between Nigeria and South Africa, the board members in Nigeria have the highest shareholdings. Outside board members are not paid but receive compensation for their activities on the firm's behalf. The average amount of board compensation to the board members of manufacturing firms in South Africa was R802,396; in Nigeria, it was ₦144,748,057; and in Ghana, it was GH¢7,358,264. The board members in Nigeria received the highest compensation, followed by the board members in Ghana, and the board members in South Africa received the lowest compensation compared to the board members in the three countries.
If the board tenure is increased by one more year, the ROIC of the firms will increase by 0.518%, 0.795%, and 0.385%, respectively.
If the number of meetings by the board is increased by 1%, there would be a fall in ROIC of 0.698%, 0.743%, and 0.230%, respectively.
the number board size is increased by 1%, there would be a fall in the ROIC by the manufacturing firms in South Africa of 0.865% but a rise of 0.295% and 0.252%, respectively, in the ROIC by the firms in Nigeria and Ghana.
If CEO duality is increased among the firms by 1%, there would be a fall in ROIC of 0.933% for manufacturing firms in South Africa, but in Nigeria and Ghana, there would be a rise in ROIC of 0.111% and 0.430%, respectively.
If the number of non-executive directors on the corporate board is increased by 1%, there would be a rise in the ROIC of the firms in South Africa and Nigeria of 0.329% and 0.015%, respectively, while the firms in Ghana would experience a fall in ROIC of 0.229%.
If the percentage of female presentations on boards is increased by 1%, there would be a rise in the ROIC of the firms in South Africa by 0.05%, while firms in Nigeria and Ghana would experience a fall of 0.171% and 0.842%, respectively.
If the percentage of shareholding by the firms is increased by 1%, there would be a fall in ROIC of 0.253% for the firms in South Africa, while there would be a rise of 0.138% and 0.412%, respectively, for the firms in Nigeria and Ghana.
If the compensation paid to the board members is increased by 1%, there would be a rise in the ROIC of the firms in South Africa by 10.8% and a fall in the ROIC of the firms in Nigeria and Ghana by 96.9% and 92.0%, respectively.
For businesses in South Africa and Nigeria, an additional year of board members would result in RONOA declines of 0.899% and 0.489%, respectively, while businesses in Ghana would see a 0.251% increase.
If the number of meetings of the board members is increased by 1%, RONOA will rise by 0.197%, 0.839%, and 0.344%, respectively
If the board size is increased by 1%, there will be a fall in the RONOA of the firms in South Africa, Nigeria, and Ghana of 0.405%, 0.401%, and 0.139%, respectively.
If CEO duality is increased by 1%, the RONOA of the firms in South Africa and Nigeria will fall by 0.595% and 0.700%, respectively, but the RONOA of the firms in Ghana will fall by 0.925%.
If the percentage of female presentations on boards is increased by 1%, the RONOA of the firms will increase by 0.454%, 0.137%, and 0.151%, respectively.
If the shareholdings of the board members are increased by 1%, RONOA for the firms in South Africa will rise by 0.14%, while that of the firms in Nigeria and Ghana will fall by 0.230% and 0.629%, respectively.
If the compensation to the board members is increased by 1%, the RONOA of the firms will rise by 26.7%, 11.5%, and 79.9%, respectively.
If the board tenure is increased by one more year, Tobin's Q for the firms will increase by 0.006%, 0.014%, and 0.042%, respectively.
If the number of meetings of the board is increased by 1%, Tobin's Q of the firms in South Africa and Ghana will increase by 0.013% and 0.105%, respectively, while there would be a fall in Tobin's Q of the firms in Nigeria by 0.050%.
If the size of the boards of the manufacturing companies in South Africa, Nigeria, and Ghana is increased by 1%, there would be a fall in Tobin's Q of 0.001%, 0.007%, and 0.049%, respectively.
If the CEO duality is increased by 1%, Tobin's Q of the firms in South Africa and Ghana will increase by 0.014% and 0.352%, respectively, but the firms in Nigeria will have a fall in Tobin's Q of 0.025%.
If the percentage of non-executive directors is increased by 1%, there would be a rise in Tobin's Q of the manufacturing firms of 0.011%, 0.192%, and 0.004%.
If the percentage of female presentation is increased by 1%, there will be a rise in Tobin's Q of the firms by 0.092%, 0.035%, and 0.011%, respectively
If the shareholdings of the board members are increased by 1%, there would be a fall in Tobin's Q of 0.003%, 0.015%, and 0.017%, respectively.
If the compensation to the board members is increased by 1%, Tobin's Q will rise by 7.5%, 1.7%, and 2.9%, respectively.
One of the objectives of the business is to grow and operate in the unforeseeable future. The management of the business operations is left in the hands of management, which sometimes has different goals from the objective of the business. Failure to regulate management activities has led to the collapse of prominent businesses in Africa, Europe, and Ghana. A key example is Enron Corporation. To ensure that the objectives of the management are aligned with those of the organization and that the management acts in the best interest of the shareholders. The introduction of corporate governance has helped to reduce corporate scandals and contributed to the improvement of firms manufacturing performance. Corporate governance and financial performance have been researched for the past five years, where ROA and ROA were used as proxies for financial performance. However, ROA and ROE are not the only proxies to measure firms' financial performance. The return on invested capital (ROIC) and return on net operating assets (RONOA) have not yet been used as proxies for financial performance to find the effect of corporate governance on them.
The purpose of the study is to find the effect of corporate governance on financial performance using ROIC, RONOA, and Tobin's Q as proxies for financial performance and board tenure, the number of board meetings, the board size, CEO duality, non-executive directors, female presentation, board ownership, and board compensation as proxies for corporate governance. The study compares the effect of corporate governance on the financial performance of three African countries: South Africa, Ghana, and Nigeria. The study used 60 manufacturing companies in all, 27 from South Africa, 17 from Nigeria, and 14 from Ghana. The data used for the study for South Africa was downloaded from Thomson Reuters Eikon DataStream, and that of Ghana and Nigeria was extracted from their annual reports. The study used a purposive sampling technique to select all the manufacturing companies between 2011 and 2020. The manufacturing companies that do not have data up to the year were not included in the study. The GMM and FMOLS were the methods used to estimate the effect of corporate governance on the financial performance of manufacturing firms.
In "References" should be mentioned sources of data collected by authors. Only in "Conclusions" the authors wrote that "the data used for the study for South Africa was downloaded from Thomson Reuters Eikon DataStream, and that of Ghana and Nigeria was extracted from their annual reports". Such an explanation should be moved to "Sample and data".
Responses
We have done so according to your recommendation.
Sample and data
The study selected three countries: South Africa, Nigeria, and Ghana. The three countries were chosen for the study based on manufacturing companies' data availability and adoption of corporate governance by the manufacturing companies listed on the stock markets. The study used companies in the manufacturing sector instead of those in the service sector. In Ghana, manufacturing contributes significantly to the country's economic growth by offering employment, paying taxes, and performing corporate social responsibilities [50]. In Nigeria, manufacturing companies contribute to the social and economic development of the country [51]. South Africa is the leading and most influential development, contributing significantly to the gross domestic product [52]. The study used a purposive sampling technique to select 29 manufacturing companies in South Africa between 2011 and 2020, giving 290 firm years of observation; in Nigeria, the study selected 17 manufacturing companies between 2011 and 2020, giving 170 firm years of observation; and in Ghana, 14 manufacturing companies between 2011 and 2020 were selected, giving 140 firm years of observation. The data used for the study from South Africa was downloaded from Thomson Reuters Eikon DataStream, and that of Ghana and Nigeria was extracted from their annual reports published. The companies selected in South Africa were listed on the Johannesburg Stock Exchange; the firms in Nigeria were listed on the Nigeria stock market; and the firms in Ghana were listed on the Ghana stock market.
The information in "Authors contribution" - "M.B.A, my supervisor, who finally approved the work" suggests that this manuscript is a short version or part of the thesis (Master's or Ph.D.). The original "work" description should be added to "References".
Response
We are so grateful for your input. M.B.A. (Murad Bein Abdurahman) is the second author who supervised and edited the work. He is not an MBA supervisor. We did the study together, and most of the content was approved by him. The study is not a thesis.
On page 10 authors wrote: “"BTE" denotes board tenure; "NBM" denotes the number of board meetings in a year; "BSZ" denotes board size; "CSY" denotes CEO duality; "NXD" denotes non-executive director; "FPN" denotes female presentation on the corporate board; "BOP" denotes board ownership; "BCN" denotes board compensation” - when 9 lines upper they wrote the same: “Board tenure – BTE, Number of board meetings – NBM […]”. Such content duplications should be reduced.
Responses; We have done so according to your recommendation. We have taken it off.
Variables |
Abbreviations |
Measurement |
Dependent variable: |
|
|
Return on invested capital |
ROIC |
Net operating profit after tax / (debt + equity) *100 |
Return on net operating assets |
RONOA |
Net operating profit after tax / net operating assets. Net operating asset = operating assets – operating liabilities |
Tobin’s Q |
TOBB |
Total Market Value of Firm/ total asset value of the firm |
Independent variable: |
|
|
Board tenure |
BTE |
The number of years the firm’s directors have served on the board / total directors serving on the board |
Number of board meetings |
NBM |
Total number of meetings in a year |
Board size |
BSZ |
Total number of board member on the corporate board |
CEO duality |
CSY |
Dummy variable if the CEO is a board chair marked ‘1’ or otherwise ‘0’ |
Non-executive director |
NXD |
Total number of non-executive directors / total number of board members |
Female presentation |
FPN |
Total number of females / total number of board members |
Board ownership |
BOP |
Total shareholding by the directors / total outstanding shares |
Board compensation |
BCN |
Log of compensation to board members |
Comments on the Quality of English Language
Poor English caused problems with a proper understanding of content. For example: "The study selected three countries for the study: South Africa, Nigeria, and Ghana. The three countries were chosen due to the availability of data and the fact that they are in Africa, where governance is practiced"; "Established countries like Germany, the United States, the United Kingdom, and others have taken effective measures to raise corporate governance standards and forestall future failures"; "According to a World Bank indicator for 2021, in Sub-Saharan Africa's total GDP, the manufacturing firms in Nigeria contributed a value-added value of 5%, South Africa contributed a value-added value of 15%, and Ghana contributed a value-added value of 12%"; "Ghana, Nigeria and South Africa are among the fastest-growing manufacturing industries in most African" and others.
Response
We are grateful for your insight. We gave the work to three English teachers who edited it, and some grammatical errors are now corrected.
The study selected three countries: South Africa, Nigeria, and Ghana. The three countries were chosen for the study based on manufacturing companies' data availability and adoption of corporate governance by the manufacturing companies listed on the stock markets.
Advanced countries like Germany, the United States, and the United Kingdom have taken adequate measures to raise corporate governance standards and prevent such failures. Several African countries, notably South Africa, Nigeria, and Ghana have instituted measures in the same regard.
According to a World Bank indicator for 2021, in Sub-Saharan Africa's total GDP, the manufacturing firms in Nigeria contributed 5%, South Africa contributed 15%, and Ghana contributed 12%.
Manufacturing companies in Ghana, Nigeria, and South Africa are among Africa's fastest-growing manufacturing industries
Submission Date
21 April 2023
Date of this review
06 May 2023 07:46:49

Reviewer 4 Report
In the study, the authors took up the important and current topic of the importance of corporate governance in shaping the economic situation of enterprises. They reviewed the literature on the subject under study (results of studies by other authors), discussed the situation in the countries studied and presented the results of their own research conducted using the method proposed by them. They adopted research objectives concerning the impact of corporate governance (features related to the functioning of the organization's board of directors) on the financial situation of production enterprises. As distinguishing features of corporate governance, they adopted eight features concerning the organization and functioning of management, including the period of work in the management board or the participation of women in this body. They developed three models to determine the impact of corporate governance on return on invested capital, RONOA and the relationship between the company's market value and the value of its assets. Empirical research was conducted using data from 60 companies from three countries: South Africa, Nigeria and Ghana. As a result of the calculations - made within three models - the authors recognized the relationships between the indicated economic results (features) (dependent variables) and the elements adopted to determine the corporate governance of the surveyed enterprises (independent variables). They presented many detailed results of various calculations for individual features. These results are interesting, although they are difficult to study. According to the reviewer, it would be better to limit the detailed commentary on the numbers presented in the tables and pay more attention to more general relationships. The weakness of the study is the lack of several synthetic conclusions from the research. A huge amount of detailed data makes the reader a bit lost in them and has to look for specific conclusions from the research himself.
In general, the study is interesting, valuable, worthy of publication in its current state, however, in order to improve it, it would be advisable in part 5 (Conclusions) to delete lines from 853 to 881, which are a repeat (for the third time) of information already presented before, and in their place to introduce some specific – generalizing, synthetic conclusions from the conducted research.
Author Response
We want to thank you for taking the time to read the manuscript and provide insight and contributions that led to shaping the paper. We are most grateful.
Some detailed commentaries have been taken out of the manuscript.
If the board tenure is increased by one more year, the ROIC of the firms will increase by 0.518%, 0.795%, and 0.385%, respectively.
If the number of meetings by the board is increased by 1%, there would be a fall in ROIC of 0.698%, 0.743%, and 0.230%, respectively.
the number board size is increased by 1%, there would be a fall in the ROIC by the manufacturing firms in South Africa of 0.865% but a rise of 0.295% and 0.252%, respectively, in the ROIC by the firms in Nigeria and Ghana.
If CEO duality is increased among the firms by 1%, there would be a fall in ROIC of 0.933% for manufacturing firms in South Africa, but in Nigeria and Ghana, there would be a rise in ROIC of 0.111% and 0.430%, respectively.
If the number of non-executive directors on the corporate board is increased by 1%, there would be a rise in the ROIC of the firms in South Africa and Nigeria of 0.329% and 0.015%, respectively, while the firms in Ghana would experience a fall in ROIC of 0.229%.
If the percentage of female presentations on boards is increased by 1%, there would be a rise in the ROIC of the firms in South Africa by 0.05%, while firms in Nigeria and Ghana would experience a fall of 0.171% and 0.842%, respectively.
If the percentage of shareholding by the firms is increased by 1%, there would be a fall in ROIC of 0.253% for the firms in South Africa, while there would be a rise of 0.138% and 0.412%, respectively, for the firms in Nigeria and Ghana.
If the compensation paid to the board members is increased by 1%, there would be a rise in the ROIC of the firms in South Africa by 10.8% and a fall in the ROIC of the firms in Nigeria and Ghana by 96.9% and 92.0%, respectively.
For businesses in South Africa and Nigeria, an additional year of board members would result in RONOA declines of 0.899% and 0.489%, respectively, while businesses in Ghana would see a 0.251% increase.
If the number of meetings of the board members is increased by 1%, RONOA will rise by 0.197%, 0.839%, and 0.344%, respectively
If the board size is increased by 1%, there will be a fall in the RONOA of the firms in South Africa, Nigeria, and Ghana of 0.405%, 0.401%, and 0.139%, respectively.
If CEO duality is increased by 1%, the RONOA of the firms in South Africa and Nigeria will fall by 0.595% and 0.700%, respectively, but the RONOA of the firms in Ghana will fall by 0.925%.
If the percentage of female presentations on boards is increased by 1%, the RONOA of the firms will increase by 0.454%, 0.137%, and 0.151%, respectively.
If the shareholdings of the board members are increased by 1%, RONOA for the firms in South Africa will rise by 0.14%, while that of the firms in Nigeria and Ghana will fall by 0.230% and 0.629%, respectively.
If the compensation to the board members is increased by 1%, the RONOA of the firms will rise by 26.7%, 11.5%, and 79.9%, respectively.
If the board tenure is increased by one more year, Tobin's Q for the firms will increase by 0.006%, 0.014%, and 0.042%, respectively.
If the number of meetings of the board is increased by 1%, Tobin's Q of the firms in South Africa and Ghana will increase by 0.013% and 0.105%, respectively, while there would be a fall in Tobin's Q of the firms in Nigeria by 0.050%.
If the size of the boards of the manufacturing companies in South Africa, Nigeria, and Ghana is increased by 1%, there would be a fall in Tobin's Q of 0.001%, 0.007%, and 0.049%, respectively.
If the CEO duality is increased by 1%, Tobin's Q of the firms in South Africa and Ghana will increase by 0.014% and 0.352%, respectively, but the firms in Nigeria will have a fall in Tobin's Q of 0.025%.
If the percentage of non-executive directors is increased by 1%, there would be a rise in Tobin's Q of the manufacturing firms of 0.011%, 0.192%, and 0.004%.
If the percentage of female presentation is increased by 1%, there will be a rise in Tobin's Q of the firms by 0.092%, 0.035%, and 0.011%, respectively
If the shareholdings of the board members are increased by 1%, there would be a fall in Tobin's Q of 0.003%, 0.015%, and 0.017%, respectively.
If the compensation to the board members is increased by 1%, Tobin's Q will rise by 7.5%, 1.7%, and 2.9%, respectively.
We took some content from the study as you suggested (lines 853 to 881) and added references to the conclusions.
One of the objectives of the business is to grow and operate in the unforeseeable future. The management of the business operations is left in the hands of management, which sometimes has different goals from the objective of the business. Failure to regulate management activities has led to the collapse of prominent businesses in Africa, Europe, and Ghana. A key example is Enron Corporation. To ensure that the objectives of the management are aligned with those of the organization and that the management acts in the best interest of the shareholders. The introduction of corporate governance has helped to reduce corporate scandals and contributed to the improvement of firms manufacturing performance. Corporate governance and financial performance have been researched for the past five years, where ROA and ROA were used as proxies for financial performance. However, ROA and ROE are not the only proxies to measure firms' financial performance. The return on invested capital (ROIC) and return on net operating assets (RONOA) have not yet been used as proxies for financial performance to find the effect of corporate governance on them.
The purpose of the study is to find the effect of corporate governance on financial performance using ROIC, RONOA, and Tobin's Q as proxies for financial performance and board tenure, the number of board meetings, the board size, CEO duality, non-executive directors, female presentation, board ownership, and board compensation as proxies for corporate governance. The study compares the effect of corporate governance on the financial performance of three African countries: South Africa, Ghana, and Nigeria. The study used 60 manufacturing companies in all, 27 from South Africa, 17 from Nigeria, and 14 from Ghana. The data used for the study for South Africa was downloaded from Thomson Reuters Eikon DataStream, and that of Ghana and Nigeria was extracted from their annual reports. The study used a purposive sampling technique to select all the manufacturing companies between 2011 and 2020. The manufacturing companies that do not have data up to the year were not included in the study. The GMM and FMOLS were the methods used to estimate the effect of corporate governance on the financial performance of manufacturing firms.
This is what we have put there now.
The effect of corporate governance on financial performance has been investigated to determine how corporate governance mechanisms affect financial performance. The study compares the effects of the corporate governance structures of the manufacturing firms in Nigeria, Ghana, and South Africa on ROIC, RONOA, and Tobin’s Q. The study results would contribute to building investors’ confidence in the governance structures of manufacturing firms in Africa.
Submission Date
21 April 2023
Date of this review
27 Apr 2023 13:24:09

Round 2
Reviewer 3 Report
Compared to the previous version of the manuscript, the authors made significant changes and improvements. However, it is worth correcting the following minor shortcomings:
1. The previous recommendation was to reduce the content of the manuscript (31 pages). Instead of this, the new version is one page longer. For example, the authors unnecessarily added “Table 1: Summary of hypothesis”. There is no need to repeat twice eight hypotheses clearly formulated in the manuscript.
2. In “References” still lack the source “Thomson Reuters Eikon DataStream”, used by authors as a source of data from South Africa and sources as “annual reports of Ghana and Nigeria”.
3. The description of previous studies on the effect of corporate governance on financial performance, published in the literature (lines 88-121) should be moved to the chart “2. Literature Review”.
4. After the sentence “There are many corporate governance theories, but the ones that support the studies are discussed below” (lines 170-171) should be mentioned only theories, like: Agency theory, Stakeholder Theory, and Resource Dependency Theory. Unnecessarily added to these theories issues such as: “Corporate Governance of South Africa”, “Corporate Governance in Nigeria”, and “Corporate Governance in Ghana” (lines 241-290) should be moved to the separate chart “Corporate governance regulations in South Africa, Nigeria, and Ghana” or similar.
Author Response
Reviewer three
Open Review
( ) I would not like to sign my review report
(x) I would like to sign my review report
Quality of English Language
( ) I am not qualified to assess the quality of English in this paper
( ) English very difficult to understand/incomprehensible
( ) Extensive editing of English language required
( ) Moderate editing of English language
( ) Minor editing of English language required
(x) English language fine. No issues detected
Yes |
Can be improved |
Must be improved |
Not applicable |
|
Is the content succinctly described and contextualized with respect to previous and present theoretical background and empirical research (if applicable) on the topic? |
( X) |
( ) |
( ) |
( ) |
Are all the cited references relevant to the research? |
(x) |
( ) |
( ) |
( ) |
Are the research design, questions, hypotheses and methods clearly stated? |
(x) |
( ) |
( ) |
( ) |
Are the arguments and discussion of findings coherent, balanced and compelling? |
(x) |
( ) |
( ) |
( ) |
For empirical research, are the results clearly presented? |
(x) |
( ) |
( ) |
( ) |
Is the article adequately referenced? |
( ) |
(x) |
( ) |
( ) |
Are the conclusions thoroughly supported by the results presented in the article or referenced in secondary literature? |
(x) |
( ) |
( ) |
( ) |
Comments and Suggestions for Authors
Compared to the previous version of the manuscript, the authors made significant changes and improvements. However, it is worth correcting the following minor shortcomings:
- The previous recommendation was to reduce the content of the manuscript (31 pages). Instead of this, the new version is one page longer. For example, the authors unnecessarily added “Table 1: Summary of hypothesis”. There is no need to repeat twice eight hypotheses clearly formulated in the manuscript.
Response:
Thank you once again for all your insight that contributed to the shaping of the paper. “Table 1: Summary of Hypotheses” was one of the recommendations by other reviewers. However, we have taken it from the manuscript. We hope it will not cause any other issues with the other reviewers.
- In “References” still lack the source “Thomson Reuters Eikon DataStream”, used by authors as a source of data from South Africa and sources as “annual reports of Ghana and Nigeria”.
Responses
We thank you so much for the insight. We have done it.
Amin & Cek [132] used data from Thomson Reuters Eikon to examine the effect of capital structure deviation from the golden ratio on the financial performance of firms in the UK and French stock markets. Boachie [25] and Musah & Adutwumwaa [24] also extracted data from the annual report of firms listed in the Ghana Stock Market to find the effect of the corporate governance system on financial performance, and in Nigeria, Urhoghide & Omolaye [22] and Yusuf, Bambale, and Abdullahi [23] also used data from firms listed on the Nigerian Stock Market for the same purpose
- The description of previous studies on the effect of corporate governance on financial performance, published in the literature (lines 88-121) should be moved to the chart “2. Literature Review”.
Response:
Please know that we have done it according to your suggestions. Thank you so much.
2.3 Empirical studies
There have been studies on the effect of corporate governance on financial per-formance in South Africa. Dzingai & Fakoya [19] used companies in the mining sector listed between 2010 and 2015 and found that board size has a negative and statistically insignificant effect on ROE, while board independence has a positive and statistically significant effect on ROE. Toshiba et al. [20] found that corporate attributes had a posi-tive and statistically significant effect on financial performance on manufacturing companies during and after the 2008 financial crisis. Karaye & Büyükkara [21] used companies listed in Johannesburg stock market data between 2013 and 2019 and found that the corporate governance index has a positive and statistically significant rela-tionship with ROA. In Nigeria, Urhoghide & Omolaye [22] used 14 oil and gas compa-nies listed in the Nigerian Stock Market between 2008 and 2015. They found that board size and female presentation on the board have positive and statistically significant ef-fects on financial performance. Yusuf, Bambale, and Abdullahi [23] used 8 financial in-stitutions listed in the Nigerian stock market in 2015 and found that board size and composition have positive and statistically insignificant effects on financial perfor-mance. In Ghana, Musah & Adutwumwaa [24] used 10 rural banks' years of data be-tween 2010 and 2019 and found that board size and CEO duality have positive and sta-tistically insignificant effects on ROA and ROE. Boachie [25] used 23 financial institu-tions’ data sets between 2006 and 2018, and the study showed that CEO duality and non-executive directors have a positive and statistically significant relationship with financial performance. Aside from South Africa, Nigeria, and Ghana, similar studies have been conducted across the globe. Hakimah et al. [26] used 50 samples of Indone-sian SMEs between 2013 and 2017 and found that board size and female presentation have a positive and statistically significant effect on ROA. Kyere & Ausloos [27] used 252 firms listed on the London Stock Exchange in the year 2014 and found that board size, independent directors, and CEO duality have a positive and statistically significant relationship with ROA, while a positive and statistically insignificant relationship was found on Tobin's Q. Al-Homaidi [28] used 30 banks between 2013 and 2016 listed on the Bombay Stock Exchange and found a statistically significant relationship with ROA. These studies used ROA, ROE, EPS, and Tobin's Q to measure financial performance. However, the effect of corporate governance on financial performance, where the return on invested capital (ROIC) and return on net operating assets (RONOA) measure financial and operating performance, has not been well explored in the literature. Also, comparative studies have not been carried out on these three countries, South Africa, Ghana, and Nigeria, to find out corporate governance's effect on manufacturing firms' financial performance
- After the sentence “There are many corporate governance theories, but the ones that support the studies are discussed below” (lines 170-171) should be mentioned only theories, like: Agency theory, Stakeholder Theory, and Resource Dependency Theory. Unnecessarily added to these theories issues such as: “Corporate Governance of South Africa”, “Corporate Governance in Nigeria”, and “Corporate Governance in Ghana” (lines 241-290) should be moved to the separate chart “Corporate governance regulations in South Africa, Nigeria, and Ghana” or similar.
Response:
We are grateful for such insight. We have done it accordingly. Thank you so much.
2.1 Theories of corporate governance
2.1.1 Agency theory
Managerial decisions in today's widely owned corporations deviate from what is necessary to maximize shareholder returns. According to agency theory, managers are agents, and owners are principals; an agency loss occurs when owners' returns on their residual claims are lower than if the owners themselves exercised direct management of the firm [32]. Mechanisms for minimizing agency losses are outlined in the agency theory. They include incentive programs for managers that provide financial rewards for maximizing shareholder value. Such strategies usually involve top executives buying shares, sometimes at a discount, to align their financial interests with shareholders [33]. Several similar systems relate CEO salary and benefits to shareholder returns and defer a portion of executive income to the future to promote long-term value maximization and discourage short-term executive activity that undermines business value [34]. When the managers of an organization have different goals from the shareholders and the management withholds essential information about manufacturing companies from the shareholders, this may impact the business's financial performance. When the share-holders of the manufacturing companies would not be willing to contribute additional funds to the corporation when they found that their objectives and information received differed from those of the managers.
2.1.2 Stewardship Theory
The stewardship theory of corporate governance downplays the possibility of tensions between business management and shareholders and favors an insider-dominated board of directors [35]. The idea presupposes that manager are trust-worthy and value their reputations highly. As a result, the market for highly regarded managers, who command higher salaries, is the essential tool to regulate behavior [36]. Financial reporting, disclosure, and auditing are still essential, but there is a fundamental assumption that these practices are necessary to confirm management's inherent integrity. One of the ways the managers account to the corporation's shareholders is through annual general meetings. When the managers did not account to the share-holders how effectively they used the corporation's assets to generate enough earnings, represented by ROA and ROE. The shareholders, through the board of directors, may not approve any assets the managers may need in the future to run the corporation, which could affect the corporation's financial performance.
2.1.3 Stakeholder Theory
According to stakeholder theory, a company or organization must serve multiple constituencies. According to stakeholder theory, everyone who impacts the company or its operations in any way is a stakeholder, including environmental organizations, clients, employees, local communities, suppliers, governmental organizations, and more [37]. According to stakeholder theory, organizations and businesses will be more successful in the long run if they prioritize their constituents' needs. According to share-holder theory, a company's primary purpose should be to advance the interests of its shareholders. Shareholder theory translates to making more profit at any cost business philosophy because shareholders are most concerned with financial growth [38]. The success of the corporation depends on its primary and secondary stakeholders. The corporation's failure to meet or address its stakeholders' needs may lead to its collapse in the future. For example, suppose the manufacturing companies fail to honor their tax obligations to the government. In that case, it may affect their operations due to the frustration they would face from the government, or the government could collapse the corporation. Therefore, stakeholder theory is vital to a manufacturing company's corporate governance and financial performance.
2.1.4 Resource Dependency Theory
According to resource dependency theory [39], the board of directors plays a pivotal role in securing the necessary external resources for an organization. Theorists of resource dependency emphasize the importance of having third-party representatives oversee the acquisition of essential resources. The availability of resources improves organizational effectiveness, productivity, and survival. The directors give the company credibility [40] and provide information, expertise, and connections to important stakeholders like government policymakers, consumers, and business partners. The smooth running of a corporation depends on its resource. Failure to supply the corporation with resources in terms of processing inputs could affect the corporation's performance. Every successful corporation depends on its inputs, such as human and capital resources.
2.2 Corporate governance regulations in South Africa, Nigeria, and Ghana
2.2.1 Corporate Governance of South Africa
South Africa uses a mix of common law, legislation, soft law, and market regulation to govern its corporations [41]. South Africa collaborates with its G20 peers to establish rules for the global regulation of financial markets. Its leadership in spreading integrated reporting worldwide has informed its approach to corporate governance, commonly referred to as the King Code, the South African Corporate Governance Code was primarily authored by the retired chairman of the International Integrated Reporting Council and the Global Reporting Initiative. Companies must adhere to various benchmark requirements set forth by the King Code in their public disclosures and in-ternal operations [42]. A general guide on how businesses should operate can help standardize sectors and attract foreign investment. All firms doing business in South Africa must adhere to some rudimentary norms of conduct, as this is one of the King Code's primary functions [43]. Shareholders and the general public can hold a firm ac-countable if it disobeys a King Code guideline.
2.2.2 Corporate Governance in Nigeria
A statutory framework and subsidiary legislation issued by the necessary regulatory authorities make up the Nigerian corporate governance regime [45]. There are two types of laws: general laws and industry-specific legislation. The sector-specific rules only apply to businesses that operate within their particular sector or industry, while the general laws apply to all entities formed in Nigeria [46]. The general laws are:
The Act Concerning Companies and Allied Matters, the Corporate Affairs Com-mission (CAC) enforces the CAMA. The Investments and Securities Act of 2007 (ISA) established the Securities and Exchange Commission (SEC) as its regulatory body. This organization is responsible for enforcing the Financial Reporting Council of Nigeria Act 2011 (FRCN Act) provisions.
The Banks and Other Financial Institutions Act (BOFIA) and the Insurance Act are two examples of industry-specific laws (IA). The CAMA is the primary law that defines the broad parameters of the Nigerian corporate governance regime [46]. Private com-panies, limited by shares or guarantees, limitless corporations, and public companies, which shares can only limit, are all described as possible forms of incorporation. The CAMA also specifies several corporate bodies' roles, responsibilities, and authority, the company's governance and management systems, and the management requirements [47]. However, the ISA lays forth the Nigerian securities market's rules and regulations. It specifies the liquidity requirements and the operational norms for market participants, operators, and stakeholders.
2.2.3 Corporate Governance in Ghana
Statutory law, subsidiary legislation, regulatory guidelines, and directives comprise most of Ghana's corporate governance framework for publicly traded companies [48]. The Corporate Governance Code is not legally binding; it serves as a guideline against which publicly traded companies and other organizations active in the securities market can be measured. Non-compliance with the strict mandatory rules that some financial services authorities have established for the governance structures and control systems of regulated organizations could have severe consequences for the companies' licenses. For instance, the National Insurance Commission has established regulations for both life and non-life insurers regarding governance and risk management. The comprehensive framework outlines the minimal standards for internal control systems and corporate governance frameworks that insurers must follow. This includes the audit and risk management functions, the board's makeup, and the roles of any relevant committees [49].
Submission Date
21 April 2023
Date of this review
16 May 2023 08:04:46
